# AUTOMATED FEATURE ENGINEERING BY PROMPTING

## ABSTRACT

Automated feature engineering (AutoFE) liberates data scientists from the burden of manual feature construction, a critical step for tabular data prediction. While the semantic information of datasets provides valuable context for feature engineering, it has been underutilized in most existing works. In this paper, we introduce AutoFE by Prompting (FEBP), a novel AutoFE algorithm that leverages large language models (LLMs) to process dataset descriptions and automatically generate features. Incorporating domain knowledge, the LLM iteratively refines feature construction through in-context learning of top-performing example features and provides semantic explanations. Our experiments on real-world datasets demonstrate the superior performance of FEBP over state-of-the-art AuoFE methods. We also conduct ablation study to verify the impact of dataset semantic information and examine the behavior of our LLM-based feature search process.

## 1 INTRODUCTION

Tabular data, a form of structured data comprising instances and attributes, have extensive use in vast domains, e.g., credit assessment, market prediction, and quality control. Traditional machine learning models, especially tree-based models (Breiman, 2001; Ke et al., 2017), have strong performance on tabular datasets of small and medium sizes (Grinsztajn et al., 2022) and good interpretability. Feature engineering is the process of computing new features from feature attributes of a dataset to enhance downstream model performance, which is crucial for traditional ML models as the new features extract useful information for target prediction by capturing complex non-linear relationships. Feature engineering by hand requires domain expertise to alleviate the significant human labor.

Automated feature engineering (AutoFE) uses high-level algorithms and models to automate the FE process such that the performance is comparable to domain experts. Existing AutoFE methods, such as (Zhu et al., 2022a;b; Zhang et al., 2023), compute and evaluate a large number of features in a trial-and-error manner. While some learn to optimize the feature utility during the FE process, these methods do not utilize domain knowledge to guide the feature search. The need to start from scratch for new datasets or downstream models hampers their effectiveness and efficiency. Besides, these methods do not offer explanation of the computed features, impairing the interpretability.

The descriptions contained in tabular datasets provide rich context for feature engineering. Domain experts consult attribute descriptions to select relevant feature attributes and compute new features useful for predicting the target. For example, the *square footage* of a house times the *average housing price per square foot* in the neighborhood can be a good predictor of the *market value* of the house. Large language models (LLMs) (Radford et al., 2019; Brown et al., 2020; OpenAI, 2023; Touvron et al., 2023a;b), pretrained on large volumes of text data, handle general natural language processing tasks and encapsulate extensive domain knowledge. Under proper prompt instructions, the LLM may process the dataset semantic information and utilize its knowledge to automatically compute features in a manner similar to domain

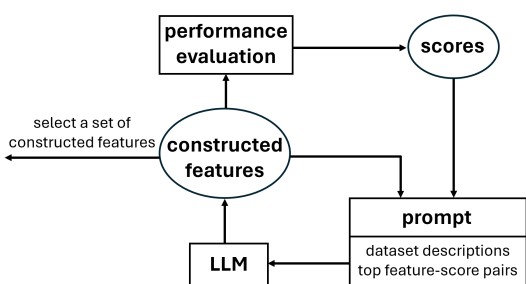

**Figure 1:** Overview of FEBP: (1) prompting the LLM to construct new features by providing dataset descriptions and example features; (2) evaluating the constructed features; (3) updating the prompt with top-performing features and scores; and (4) selecting a set of features and adding them to the dataset.

experts. The work by Hollmann et al. (2023) demonstrates the potential of applying LLMs for AutoFE, but it is not sufficiently effective in terms of feature search.

We propose AutoFE by Prompting (FEBP), a novel AutoFE algorithm that leverages LLMs for effective, efficient, and interpretable feature engineering, as illustrated in Figure 1. By providing dataset descriptions and example features in canonical Reverse Polish Notation (cRPN), we prompt the LLM to generate new features. After evaluating the features, we update the prompt with top-performing features and their evaluation scores and instruct the LLM to construct further features. Through this iterative process, the LLM explores the feature space and improves solutions through in-context learning of successful examples. The semantic information of data not only guides the feature search, but helps the LLM understand and learn from the patterns in example features. Applying domain knowledge, the LLM generates semantically meaningful features and explains their usefulness. Experiments on seven real-world datasets demonstrate that FEBP significantly outperforms state-of-the-art baselines, achieving over 5% performance gain on average across three downstream models. Additionally, our ablation study shows that incorporating dataset semantic context improves the performance. We also analyze the behavior of the LLM-based feature search process and examine the effects of hyperparameters.

Our main contributions are: (1) We introduce a novel LLM-based AutoFE algorithm that utilizes dataset semantic information for automated feature search. This is the first method capable of generating features in string representations while providing semantic explanations. (2) We benchmark the performance of our approach against state-of-the-art baselines using both GPT-3.5 and GPT-4. (3) We investigate the impact of semantic context on our approach, analyze the behavior of the LLM-based feature search process, and examine hyperparameter effects.

## 2 RELATED WORK

**Large Language Models.** LLMs are large-scale general-purpose neural networks pretrained on large corpora of raw text data for natural language processing, typically built with transformer-based architectures (Vaswani et al., 2017). Generative LLMs, such as the GPT family (Radford et al., 2019; Brown et al., 2020; OpenAI, 2023) and the LLaMA family (Touvron et al., 2023a;b), are pretrained to successively predict the next token given the input text and can be finetuned using reinforcement learning from human feedback (Ziegler et al., 2019; Ouyang et al., 2022). By this means, they acquire the knowledge about syntax and semantics of human languages and are able to achieve state-of-the-art performance on various tasks like text generation, summarization, and question answering. LLMs can be adapted to specific tasks without changing model parameters through prompt engineering. Few-shot learning (Brown et al., 2020) includes examples in the prompt for the language model to learn in-context. Leveraging such capability, the LLM may function as a problem solver (Yang et al., 2024) that iteratively improves candidate solutions according to the task description and feedback. Chain-of-though (Wei et al., 2022; Kojima et al., 2022) enables complex reasoning capabilities of LLMs through intermediate reasoning steps.

**Automated Feature Engineering.** AutoFE computes new features for the input data and augments or replaces portions of the existing features, to enhance the performance of downstream models. Common AutoFE approaches include expansion-reduction (Kanter & Veeramachaneni, 2015; Horn et al., 2020; Zhang et al., 2023), evolutionary algorithms (Smith & Bull, 2005; Zhu et al., 2022a), and reinforcement learning (Khurana et al., 2018; Li et al., 2023; Wang et al., 2023). DIFER (Zhu et al., 2022b) utilizes neural networks to learn the utility of constructed features and optimize features in the embedding space. OpenFE (Zhang et al., 2023) introduces a feature boost algorithm to speedup feature evaluation. Nonetheless, these approaches do not utilize the semantic information of data, which impedes the performance and interpretability of engineered features.

**AutoFE with Domain Knowledge.** The benefits of incorporating domain knowledge in AutoFE include: (1) improving the effectiveness; and (2) reducing the cost of learning an AutoFE model, especially the feature evaluation overhead. Prior works take different directions. One direction is to transfer the knowledge through pretraining. LFE (Nargesian et al., 2017) represents features with quantile sketches transferable across datasets and inputs them to a feature transformation recommendation model. FETCH (Li et al., 2023) is an RL-based AutoFE framework taking tabular data as the state and generalizable to new data. E-AFE (Wang et al., 2023) pretrains a feature evaluator

to efficiently learn the RL-based AutoFE model. The other direction is to leverage the semantic information of datasets. KAFE (Galhotra et al., 2019) leverages knowledge graphs to identify semantically informative features relevant to the prediction task. CAAFE (Hollmann et al., 2023) manipulates Pandas data frames using the code generated from the LLM based on dataset descriptions. In our work, we adopt a compact form of feature representation in strings with pre-defined transformation operators. Our approach reduces the search space and helps the LLM learn the patterns of useful features, leading to stronger and more robust performance. We further discuss the differences between our approach and CAAFE in Appendix G.

## 3 NOTATIONS

We denote a tabular dataset as $D = \langle \mathbb{X}, \boldsymbol{y} \rangle$, where $\mathbb{X} = \{\boldsymbol{x}_1, \ldots, \boldsymbol{x}_d\}$ is the set of raw features with $\boldsymbol{x}_i \in \mathbb{R}^n$ for $i = 1, \ldots, d$ and $\boldsymbol{y} \in \mathbb{R}^n$ is the target. We construct a new feature $\tilde{\boldsymbol{x}} = t(\boldsymbol{x}_{j_1}, \ldots, \boldsymbol{x}_{j_o})$ by transforming existing features $\boldsymbol{x}_{j_1}, \ldots, \boldsymbol{x}_{j_o}$ via some operator $t \in \mathbb{R}^n \times \ldots \times \mathbb{R}^n \to \mathbb{R}^n$ of arity $o$. Given a set of transformation operators $\mathbb{T}$, we define the feature space $\mathbb{X}_\mathbb{T}$ recursively as: for any $\tilde{\boldsymbol{x}} \in \mathbb{X}_\mathbb{T}$, either $\tilde{\boldsymbol{x}} \in \mathbb{X}$; or $\exists t \in \mathbb{T}$, s.t., $\tilde{\boldsymbol{x}} = t(\tilde{\boldsymbol{x}}_{j_1}, \ldots, \tilde{\boldsymbol{x}}_{j_o})$, where $\tilde{\boldsymbol{x}}_{j_1}, \ldots, \tilde{\boldsymbol{x}}_{j_o} \in \mathbb{X}_\mathbb{T}$. To measure feature complexity, we compute the order of a feature $\tilde{\boldsymbol{x}} \in \mathbb{X}_\mathbb{T}$ as:

$$\alpha(\tilde{\boldsymbol{x}}) = \begin{cases} 0 & \text{if } \tilde{\boldsymbol{x}} \in \mathbb{X}, \\ 1 + \max_j \alpha(\tilde{\boldsymbol{x}}_j) & \text{if } \tilde{\boldsymbol{x}} = t(\tilde{\boldsymbol{x}}_{j_1}, \ldots, \tilde{\boldsymbol{x}}_{j_o}) \text{ for some } t \in \mathbb{T}. \end{cases} \quad (1)$$

The constrained feature space with the order upper bounded by $k$ is denoted as $\mathbb{X}_\mathbb{T}^{(k)} = \{\tilde{\boldsymbol{x}} \in \mathbb{X}_\mathbb{T} \mid \alpha(\tilde{\boldsymbol{x}}) \le k\}$.

We denote the performance of a downstream machine learning model algorithm $M$ on the dataset as $\mathcal{E}_M(\mathbb{X}, \boldsymbol{y})$. Our objective of AutoFE is to construct a set of features $\tilde{\mathbb{X}}^*$ to optimize the model performance by adding them to the dataset, specifically:

$$\tilde{\mathbb{X}}^* = \arg\max_{\tilde{\mathbb{X}} \subset \mathbb{X}_\mathbb{T}} \mathcal{E}_M(\mathbb{X} \cup \tilde{\mathbb{X}}, \boldsymbol{y}). \quad (2)$$

## 4 METHODOLOGY

In this section, we present AutoFE by Prompting (FEBP), a novel AutoFE algorithm leveraging the power of LLMs, particularly, the GPT models (Radford et al., 2019; Brown et al., 2020; OpenAI, 2023). The high-level idea is to provide the LLM with descriptive information of the dataset in the prompt and guide it to search for effective features using examples.

We represent features in a compact form in our prompt. A feature $\tilde{\boldsymbol{x}} \in \mathbb{X}_\mathbb{T}$ is expressible as a tree, where the leaf nodes are raw features and the internal nodes are operators. However, the expression trees of features containing commutative operators (like addition and multiplication) are not unique since the child nodes of these operators are unordered. We introduce a canonicalization scheme: arranging operator nodes before feature nodes for left skewness and lexicographically sorting the nodes within each group. We

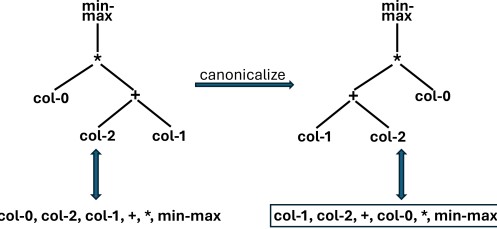

**Figure 2:** We obtain the canonical RPN (cRPN) by reordering nodes of the expression tree.

then serialize the canonical expression tree into the postorder depth-first traversal string, i.e., canonical reverse Polish notation (cRPN), ensuring the one-to-one mapping between features and string representations. We denote the feature corresponding to an RPN string $f$ as $\tilde{\boldsymbol{x}}_f$ and the set of features corresponding to a set of RPN strings $\mathbb{F}$ as $\tilde{\mathbb{X}}_\mathbb{F}$. We make further discussion in Appendix A.

Our prompt contains: (1) a meta description of the dataset (optional); (2) an indexed list of the dataset attributes, with attribute types, value ranges, and descriptions; (3) lists of transformation operators with descriptions, grouped by the arity; (4) a ranked list of example features with performance evaluation scores; and (5) an output template for new features and explanations. Figure 3 outlines the structure of our prompt. The descriptions of the dataset and attributes provide contextual information for the LLM to understand the data and apply domain knowledge. The value ranges of attributes

---

(1) Dataset description:

This dataset contains information on default payments, demographic factors, credit data, and history of payment of credit card clients ...

(2) Dataset contains the following columns:

col-0 (int) [10000, 800000]: LIMIT_BAL: Amount of given credit in NT dollars (includes individual and family/supplementary credit

col-1 (category) {1, 2}: SEX: Gender (1=male, 2=female) ...

col-23 (category) {0, 1}: default.payment.next.month: Default payment (1=yes, 0=no)

(3) We have the following unary operators:

sqrt_abs: taking the square root of the absolute value ...

We have the following binary operators:

+: summing two columns ...

Feature strings are reverse Polish notation (RPN) expressions that operate on the columns of our dataset. Each feature string constructs an extra column that is useful for the downstream model Random Forests to predict the target col-23. The model will be trained on the dataset with the constructed columns and evaluated on a holdout set. The best columns will be selected.

(4) Below are feature strings arranged in ascending order based on their performance scores. Higher scores are better.

Feature

col-17,col-21,*,col-20,+,sqrt_abs

Score

0.0011 ...

Feature

col-4,col-6,*,col-12,col-16,-,sqrt_abs,*

Score

0.0014

Give me a new feature string that is different from all strings above and has a higher score. Use no more than five operators. Make sure all columns and operators exist and do not include the target column. Follow the syntax of RPN.

(5) Output format:

Feature

(Feature name and description)

Usefulness

(Explanation why this adds useful real world knowledge to predict the target col-23 according to dataset description)

**Figure 3:** Prompt template. Sections requiring dataset information are highlighted in blue. The ranked list of feature examples and scores that gets updated in our feature search algorithm is highlighted in orange.

---

Feature

col-11,col-11,col-12,-,log,*

This feature calculates the log of the difference between the September bill statement (col-11) and the August bill statement (col-12), then multiplies it by the September bill amount.

Usefulness

This feature captures the change in the bill amount from August to September in a logarithmic scale, which can effectively highlight significant changes in spending patterns. Large fluctuations in credit card bills could be indicative of financial distress, which may impact the likelihood of defaulting on payments, while the logarithmic transformation allows for handling potential skewness in the data distribution. By incorporating this feature, the model can better understand how temporal changes in spending behavior relate to the probability of default, providing richer contextual information beyond static features.

**Figure 4:** The LLM constructs a new feature in RPN and explains its usefulness from the semantic perspective.

are useful for selecting appropriate feature transformations, e.g., min-max normalization when the scale is too large. We include the descriptions of transformation operators as they help the LLM parse example features in RPN syntax and construct syntactically valid feature strings. The output template not only structures the output but instructs the LLM to reason about the usefulness of the proposed features and offer semantic explanations, utilizing the chain-of-thought technique (Wei et al., 2022; Kojima et al., 2022). We additionally add a constraint instruction to use no more than a certain number of operators, which reduces the search space and regularizes the solutions. Figure 4 shows an example LLM output. The prompt can further include attribute statistics like mean, standard deviation, and skewness, and we leave that for future work.

We initialize the prompt with $k$ random features from the constrained feature space $\tilde{\boldsymbol{x}}_1, \ldots, \tilde{\boldsymbol{x}}_k \in \mathbb{X}_{\mathbb{T}}^{(2)}$ represented in cRPN for demonstration. This lets the LLM start search from a small feature space where it is easier to identify the basic patterns of promising features. Optionally, we can import external example features. We prompt the LLM to construct a fixed number of $m$ new feature in an iteration. For each constructed feature string $f$, we first try to obtain the cRPN expression $f^c$ to check whether $f^c$ is syntactically valid and not a duplicate of existing features. If both criteria are met, we evaluate the performance score of adding the single feature to the dataset $s = \mathcal{E}_M(\mathbb{X} \cup \{\tilde{\boldsymbol{x}}_{f^c}\}, \boldsymbol{y})$ through cross validation on the training data and add $\langle f^c, s \rangle$ to the candidate set $\mathbb{F}_{cand}$. When $f^c$ is among the top-$k$ candidate features in terms of the score $s$, we update prompt examples with the top-$k$ pairs $\langle f', s' \rangle \in \mathbb{F}_{cand}$ ranked in the ascending order, taking score increment $s' - \mathcal{E}_M(\mathbb{X}, \boldsymbol{y})$ from the baseline. We then instruct the LLM to construct additional features using the updated prompt. To select candidate features, we successively add candidate features to the dataset from the best to the worst and determine the optimal number of features to add based on validation performance, which is evaluated over sets of candidate features and thus takes feature interactions into account. Algorithm 1 summarizes our methodology. The size of the prompt scales linearly with the number of features in the dataset $d$ and the number of example features $k$ and stays roughly constant across feature construction iterations. Thus, the cost of an LLM generation step in line 3 is almost constant.

---

**Algorithm 1:** AutoFE by Prompting

---

**Input** : Dataset $D = \langle \mathbb{X}, \boldsymbol{y} \rangle$, downstream model $M$, large language model $LLM$, and optionally an external set of features with evaluation scores $\mathbb{F}_{ext}$

**Output:** A set of engineered features $\mathbb{F}$

1  Initialize prompt $P$ with dataset descriptions and example features; $\mathbb{F}_{cand} \leftarrow \mathbb{F}_{ext}$ if $\mathbb{F}_{ext}$ is available, otherwise $\mathbb{F}_{cand} \leftarrow \emptyset$; $\mathbb{F}_{set} \leftarrow \emptyset$

2  **repeat**

3     $\mathbb{F}_{LLM} = \{f_1, \ldots, f_m\} \leftarrow LLM(P)$         ▷ Feature generation

4     **for** each $f \in \mathbb{F}_{LLM}$ **do**

5         $f^c \leftarrow$ Canonicalize $f$

6         **if** $f^c$ is valid and $f^c \notin \mathbb{F}_{cand}$ **then**         ▷ Feature evaluation

7             Evaluate cross validation performance score $s \leftarrow \mathcal{E}_M(\mathbb{X} \cup \{\boldsymbol{x}_{f^c}\}, \boldsymbol{y})$ on training data

8             $\mathbb{F}_{cand} \leftarrow \mathbb{F}_{cand} \cup \{\langle f^c, s \rangle\}$

9         **end**

10    **end**

11    Update $P$ such that $P$ contains the top-$k$ $\langle f', s' \rangle \in \mathbb{F}_{cand}$ as ordered by $s'$

12    **if** feature selection **then**

13        **for** $n \leftarrow 1$ **to** $|\mathbb{F}_{cand}|$ **do**         ▷ Feature selection

14            $\mathbb{F}_n \leftarrow$ The top-$n$ features in $\mathbb{F}_{cand}$ as ordered by $s$

15            Evaluate performance score $s_n \leftarrow \mathcal{E}_M(\mathbb{X} \cup \tilde{\mathbb{X}}_{\mathbb{F}_n}, \boldsymbol{y})$ on validation data

16        **end**

17        $\mathbb{F}_{set} \leftarrow \mathbb{F}_{set} \cup \{\langle \mathbb{F}_{n^*}, s_{n^*} \rangle\}$, where $n^* \leftarrow \arg\max_n s_n$

18    **end**

19  **until** stopping criteria are met

20  **return** $\mathbb{F}$ in $\mathbb{F}_{set}$ with the maximum validation score

---

The computation cost of feature evaluation in line 7 is also constant, preserving the efficiency and scalability of our algorithm. The evaluations in line 7 and at lines 13-16 are parallelizable.

Methodologically, we instruct the LLM to act as a problem solver (Yang et al., 2024) within our algorithm. Similar to evolutionary algorithms (Smith & Bull, 2005; Zhu et al., 2022a; Morris et al., 2024) that generate new solutions through crossover and mutations on high-fitness candidates, we maintain a pool of top-performing candidate features as examples. By learning examples and scores in-context (Brown et al., 2020), the LLM can recognize patterns of promising features and propose new features that are likely to be effective. For instance, it may make analogies to, modify, or combine some example features in the prompt (Appendix F.3). Early in the search, we expect greater exploration due to the diversity of initial examples. As iterations progress, the LLM focuses more on exploiting promising feature spaces, gradually refining the search until convergence. The dataset semantic information serves as a prior that guides the selection of feature attributes and operators to enhance the effectiveness of feature search. The sampling temperature of the LLM can be adjusted to balance exploration and exploitation, with higher temperatures encouraging more diverse solutions and lower temperatures favoring incremental changes to existing examples.

We adopt the same set of transformation operators $\mathbb{T}$ as those in (Zhu et al., 2022b), including:

- Unary transformations: logarithm, reciprocal, square root, and min-max normalization;
- Binary transformations: addition, subtraction, multiplication, division, and modulo.

When computing min-max normalization, we take the minimum and maximum from the training data. Other transformations require only the information of an individual example. Hence, all our transformation operations can be performed instance by instance on each individual test example without leaking the information of other test examples. As discussed by Overman et al. (2024), data leakage is an issue that has not been properly addressed in many existing AutoFE works.

## 5 EXPERIMENTS

### 5.1 EXPERIMENTAL SETUP

We benchmark performance on seven public real-world datasets from Kaggle and UCI repositories covering different domains. The descriptive information of datasets and attributes is retrieved

**Table 1:** Dataset statistics. The selected datasets cover different domains and vary in size. Four of them are for regression tasks and three for classification tasks.

| Name | Task | # Samples | # Features | # Numerical | # Categorical |
|------|------|-----------|-----------|-------------|---------------|
| Airfoil (AF) | Regression | 1,503 | 5 | 5 | 0 |
| Boston Housing (BH) | Regression | 506 | 13 | 12 | 1 |
| Bikeshare (BS) | Regression | 731 | 10 | 6 | 4 |
| Wine Quality Red (WQR) | Regression | 1,599 | 11 | 11 | 0 |
| AIDS Clinical Trials (ACT) | Classification | 2,139 | 23 | 9 | 14 |
| Credit Default (CD) | Classification | 30,000 | 23 | 14 | 9 |
| German Credit (GC) | Classification | 1,000 | 20 | 10 | 10 |

from the sources without further processing. The downstream models we evaluate include linear models (LASSO for regression tasks and logistic regression for classification tasks), Random Forests (Breiman, 2001), and LightGBM (Ke et al., 2017). For linear models, we target-encode categorical features and min-max scale all features. We tune downstream model parameters by randomized search prior to and post AutoFE, because the model algorithm may need reconfiguration to accommodate the added features. Data are randomly split into training (64%), validation (16%), and test (20%) sets. We evaluate regression performance with $1 - (relative\,absolute\,error)$[1] and classification performance with accuracy. A higher evaluation score indicates better performance.

We compare our FEBP with the following state-of-the-art AutoFE methods:

- DIFER (Zhu et al., 2022b): A neural network-based method that optimizes features in the embedding space utilizing LSTMs to encode and decode features;

- OpenFE (Zhang et al., 2023): An expansion-reduction method that evaluates and ranks features up to a certain order using a feature boost algorithm;

- CAAFE (Hollmann et al., 2023): An LLM-based method that produces Python code based on dataset descriptions to manipulate Pandas data frames.

We employ *gpt-3.5-turbo-0125*[2] and *gpt-4-0613*[2] as the LLMs in our experiments. For FEBP, we include $k = 10$ example features in the prompt and set the temperature of GPT models to 1 based on validation performance. We prompt the LLM to construct $m = 1$ feature in each generation step for more accurate control of feature generation. We perform feature selection each time 10 new candidate features are constructed and terminate the algorithm once we have 200 candidate features. Parameters of the baseline AutoFE methods are initialized per the corresponding papers. For CAAFE (Hollmann et al., 2023), we raise the number of iterations from 10 to 20. Drastically increasing this limit causes failures due to the context window size of GPT models. We report results from five repeated runs unless stated otherwise.

## 5.2 PERFORMANCE COMPARISON

Table 2 compares the performance between FEBP and baseline methods. While there is no single method that dominates all test cases, FEBP achieves the best mean performance score and the lowest mean rank. FEBP yields over 5% average performance gain over downstream models using raw features, with over 15% gain for linear models and around 2% gain for Random Forests and LightGBM. Greater performance gain is observed using linear models because Random Forests and LightGBM can model complex non-linear relationships themselves. The Friedman-Nemenyi test shows that the performance difference between FEBP and baseline methods other than DIFER is statistically significant at the $p = 0.01$ level. We note that the post-AutoFE parameter tuning improves the performance of DIFER the most, as DIFER adds many more features to the datasets (Appendix D.8). FEBP is considerably more efficient as it evaluates only 200 candidate features during feature search, whereas DIFER evaluates over 2000 candidate features (Appendix D.9).

Additionally, we observe that the performance of FEBP or CAAFE with GPT-4 is not significantly different from that with GPT-3.5. On FEBP, GPT-4 yields better performance for linear models but slightly worse performance for Random Forests. We speculate that the stronger in-context learning capability of GPT-4 increases the likelihood of overfitting to the learning samples.

---

[1] $1 - \frac{\sum_i |y_i - \hat{y}_i|}{\sum_i |y_i - \bar{y}|}$, where $y$ is the target and $\hat{y}$ is the prediction.

[2] https://platform.openai.com/docs/models

**Table 2:** Summary of experimental results. For each compared method, the left and right columns show the results without and with parameter tuning of the downstream model algorithm post AutoFE, respectively. The best results are highlighted in boldface, and the second best results are underlined.

| Model | Dataset | Raw | DIFER | | OpenFE | | CAAFE | | | | FEBP (ours) | | | |
|---|---|---|---|---|---|---|---|---|---|---|---|---|---|---|
| | | | | | | | GPT-3.5 | | GPT-4 | | GPT-3.5 | | GPT-4 | |
| Linear Model | AF | 0.3474 | 0.5870 | 0.6090 | 0.4300 | 0.4303 | 0.4011 | 0.4016 | 0.4376 | 0.4378 | 0.6612 | 0.6616 | **0.6649** | 0.6647 |
| | BH | 0.3776 | 0.5013 | 0.4994 | 0.3900 | 0.3880 | 0.4788 | 0.4765 | 0.4503 | 0.4506 | 0.4995 | 0.5025 | 0.5184 | **0.5289** |
| | WQR | 0.2696 | 0.2475 | 0.2630 | 0.2713 | 0.2736 | 0.2742 | 0.2757 | **0.2776** | 0.2776 | 0.2722 | 0.2745 | 0.2713 | 0.2748 |
| | ACT | 0.8505 | 0.8715 | **0.8799** | 0.8729 | 0.8729 | 0.8519 | 0.8514 | 0.8565 | 0.8570 | 0.8729 | 0.8794 | 0.8766 | 0.8762 |
| | CD | 0.8267 | 0.8273 | 0.8280 | 0.8265 | 0.8268 | 0.8265 | 0.8267 | 0.8238 | 0.8238 | 0.8282 | 0.8282 | 0.8288 | **0.8288** |
| | GC | 0.7100 | 0.7140 | 0.7420 | 0.7320 | 0.7280 | 0.7350 | 0.7330 | 0.7210 | 0.7210 | 0.7570 | 0.7460 | **0.7590** | 0.7420 |
| | Mean | 0.5636 | 0.6248 | 0.6369 | 0.5871 | 0.5866 | 0.5946 | 0.5941 | 0.5945 | 0.5946 | 0.6485 | 0.6487 | **0.6532** | 0.6526 |
| Mean Rank | | 12.00 | 8.17 | 5.50 | 9.25 | 8.50 | 8.67 | 8.17 | 8.83 | 8.17 | 4.75 | 3.17 | 3.00 | **2.83** |
| Random Forests | AF | 0.7677 | 0.7650 | 0.7786 | 0.7579 | 0.7682 | 0.7711 | 0.7693 | 0.7696 | 0.7720 | 0.7709 | **0.7787** | 0.7681 | 0.7749 |
| | BH | 0.5426 | **0.5718** | 0.5701 | 0.5658 | 0.5620 | 0.5556 | 0.5556 | 0.5512 | 0.5492 | 0.5549 | 0.5533 | 0.5543 | 0.5522 |
| | BS | 0.9446 | 0.9865 | 0.9871 | 0.9901 | 0.9901 | **0.9916** | **0.9916** | 0.9818 | 0.9816 | 0.9873 | 0.9881 | 0.9845 | 0.9848 |
| | WQR | 0.3662 | 0.3838 | 0.3832 | 0.3753 | 0.3729 | 0.3718 | 0.3718 | 0.3693 | 0.3693 | **0.3862** | 0.3845 | 0.3810 | 0.3810 |
| | ACT | 0.8808 | 0.8897 | 0.8897 | 0.8832 | 0.8841 | 0.8827 | 0.8855 | 0.8827 | 0.8827 | **0.8925** | 0.8921 | 0.8893 | 0.8864 |
| | CD | 0.8293 | 0.8285 | 0.8291 | 0.8287 | 0.8285 | 0.8291 | 0.8289 | 0.8294 | 0.8287 | 0.8295 | 0.8294 | **0.8295** | 0.8276 |
| | GC | 0.7450 | 0.7550 | 0.7500 | 0.7650 | 0.7570 | **0.7690** | 0.7620 | 0.7660 | 0.7630 | 0.7640 | 0.7620 | 0.7680 | 0.7680 |
| | Mean | 0.7252 | 0.7400 | 0.7411 | 0.7380 | 0.7376 | 0.7387 | 0.7378 | 0.7357 | 0.7352 | 0.7408 | **0.7412** | 0.7392 | 0.7393 |
| Mean Rank | | 11.57 | 7.29 | 5.14 | 7.07 | 7.64 | 5.71 | 6.93 | 8.43 | 9.79 | **4.14** | 4.29 | 6.00 | 7.00 |
| Light-GBM | AF | 0.8375 | 0.8285 | 0.8411 | 0.8188 | 0.8244 | 0.8364 | 0.8348 | **0.8430** | 0.8426 | 0.8311 | 0.8392 | 0.8366 | 0.8395 |
| | BH | 0.5537 | 0.5607 | 0.5636 | **0.5693** | 0.5618 | 0.5540 | 0.5571 | 0.5478 | 0.5501 | 0.5619 | 0.5644 | 0.5642 | 0.5595 |
| | BS | 0.9429 | 0.9763 | 0.9786 | 0.9751 | 0.9797 | 0.9555 | 0.9565 | 0.9449 | 0.9487 | 0.9737 | 0.9754 | 0.9801 | **0.9813** |
| | WQR | 0.3825 | 0.4145 | **0.4182** | 0.3898 | 0.3884 | 0.4131 | 0.4035 | 0.3902 | 0.3952 | 0.4118 | 0.4171 | 0.4021 | 0.4042 |
| | ACT | 0.8832 | 0.8794 | 0.8827 | 0.8808 | 0.8799 | 0.8822 | 0.8860 | 0.8827 | 0.8818 | 0.8888 | **0.8925** | 0.8902 | **0.8925** |
| | CD | 0.8300 | 0.8283 | 0.8277 | 0.8293 | 0.8287 | 0.8296 | 0.8298 | 0.8301 | 0.8294 | 0.8301 | 0.8297 | **0.8303** | 0.8294 |
| | GC | 0.7250 | 0.7650 | 0.7600 | 0.7550 | 0.7550 | 0.7490 | 0.7550 | 0.7450 | 0.7720 | 0.7680 | 0.7720 | **0.7760** | 0.7700 |
| | Mean | 0.7364 | 0.7504 | 0.7531 | 0.7454 | 0.7476 | 0.7457 | 0.7461 | 0.7405 | 0.7457 | 0.7522 | 0.7558 | **0.7542** | 0.7538 |
| Mean Rank | | 9.43 | 8.29 | 5.86 | 8.86 | 8.57 | 8.43 | 7.71 | 8.29 | 7.93 | 5.71 | **3.57** | **3.57** | 4.79 |
| Mean | | 0.6806 | 0.7091 | 0.7140 | 0.6953 | 0.6958 | 0.6979 | 0.6976 | 0.6950 | 0.6967 | 0.7171 | 0.7185 | **0.7187** | 0.7183 |
| Mean Rank | | 10.95 | 7.90 | 5.50 | 8.35 | 8.23 | 7.55 | 7.58 | 8.50 | 8.65 | 4.88 | **3.70** | 4.25 | 4.98 |

**Table 3:** Performance comparison of FEBP with and without semantic blinding. For each compared version, the left and middle columns show the results without and with parameter tuning of the downstream model algorithm post AutoFE, respectively, and the right column shows the number of LLM responses. The results where the full version outperforms the blinded version are highlighted in boldface.

| Model | Dataset | Raw | GPT-3.5 | | | | | | GPT-4 | | | | | |
|---|---|---|---|---|---|---|---|---|---|---|---|---|---|---|
| | | | Blinded | | | Full | | | Blinded | | | Full | | |
| Linear Model | AF | 0.3474 | 0.6613 | 0.6602 | 450.0 | 0.6612 | **0.6616** | 339.8 | 0.6678 | 0.6672 | 275.0 | 0.6649 | 0.6647 | 371.4 |
| | BH | 0.3776 | 0.4678 | 0.4794 | 438.0 | **0.4995** | **0.5025** | 378.6 | 0.4869 | 0.4996 | 295.6 | **0.5184** | **0.5289** | 335.4 |
| | WQR | 0.2696 | 0.2643 | 0.2733 | 442.8 | **0.2722** | **0.2745** | 328.4 | 0.2645 | 0.2702 | 244.6 | **0.2713** | **0.2748** | 312.6 |
| | ACT | 0.8505 | 0.8790 | 0.8799 | 442.8 | 0.8729 | 0.8794 | 372.2 | 0.8720 | 0.8729 | 238.8 | **0.8766** | **0.8762** | 377.4 |
| | CD | 0.8267 | 0.8283 | 0.8283 | 454.8 | 0.8282 | 0.8282 | 342.0 | 0.8282 | 0.8289 | 238.2 | **0.8288** | 0.8288 | 250.4 |
| | GC | 0.7100 | 0.7460 | 0.7390 | 432.2 | **0.7570** | **0.7460** | 379.0 | 0.7430 | 0.7410 | 231.2 | **0.7590** | **0.7420** | 310.6 |
| | Mean | 0.5636 | 0.6411 | 0.6433 | 443.4 | **0.6485** | **0.6487** | 356.7 | 0.6437 | 0.6461 | 253.9 | **0.6532** | **0.6526** | 326.3 |
| Random Forests | AF | 0.7677 | 0.7644 | 0.7743 | 425.2 | **0.7709** | **0.7787** | 393.2 | 0.7610 | 0.7690 | 274.2 | **0.7681** | **0.7749** | 314.2 |
| | BH | 0.5426 | 0.5483 | 0.5483 | 479.2 | **0.5549** | **0.5533** | 374.4 | 0.5507 | 0.5491 | 238.4 | **0.5543** | **0.5522** | 278.6 |
| | BS | 0.9446 | 0.9628 | 0.9628 | 510.0 | **0.9873** | **0.9881** | 386.8 | 0.9535 | 0.9543 | 247.4 | **0.9845** | **0.9848** | 255.0 |
| | WQR | 0.3662 | 0.3749 | 0.3738 | 461.4 | **0.3862** | **0.3845** | 362.6 | 0.3666 | 0.3674 | 253.0 | **0.3810** | **0.3810** | 283.2 |
| | ACT | 0.8808 | 0.8864 | 0.8841 | 475.8 | **0.8925** | **0.8921** | 357.6 | 0.8874 | 0.8841 | 222.4 | **0.8893** | **0.8864** | 424.0 |
| | CD | 0.8293 | 0.8283 | 0.8282 | 497.0 | **0.8295** | **0.8294** | 349.8 | 0.8291 | 0.8286 | 375.2 | **0.8295** | 0.8276 | 304.0 |
| | GC | 0.7450 | 0.7630 | 0.7580 | 459.2 | **0.7640** | **0.7620** | 368.2 | 0.7510 | 0.7490 | 229.6 | **0.7680** | **0.7680** | 471.8 |
| | Mean | 0.6806 | 0.7326 | 0.7328 | 472.5 | **0.7408** | **0.7412** | 370.4 | 0.7285 | 0.7288 | 262.9 | **0.7392** | **0.7393** | 333.0 |
| Light-GBM | AF | 0.8375 | 0.8304 | 0.8356 | 479.6 | **0.8311** | **0.8392** | 380.2 | 0.8185 | 0.8266 | 284.6 | **0.8366** | **0.8395** | 360.6 |
| | BH | 0.5537 | 0.5503 | 0.5467 | 490.8 | **0.5619** | **0.5644** | 342.0 | 0.5500 | 0.5609 | 238.4 | **0.5642** | 0.5595 | 345.6 |
| | BS | 0.9429 | 0.9693 | 0.9691 | 480.2 | **0.9737** | **0.9754** | 380.0 | 0.9539 | 0.9536 | 312.6 | **0.9801** | **0.9813** | 236.8 |
| | WQR | 0.3825 | 0.4087 | 0.4151 | 493.0 | **0.4118** | **0.4171** | 322.8 | 0.4057 | 0.4050 | 246.8 | 0.4021 | 0.4042 | 293.6 |
| | ACT | 0.8832 | 0.8864 | 0.8883 | 513.0 | **0.8888** | **0.8925** | 367.4 | 0.8813 | 0.8748 | 229.0 | **0.8902** | **0.8925** | 359.6 |
| | CD | 0.8300 | 0.8284 | 0.8292 | 490.8 | **0.8301** | **0.8297** | 352.2 | 0.8295 | 0.8299 | 218.6 | **0.8303** | 0.8294 | 371.2 |
| | GC | 0.7250 | 0.7620 | 0.7620 | 482.4 | **0.7680** | **0.7720** | 376.6 | 0.7550 | 0.7550 | 225.0 | **0.7760** | **0.7700** | 382.2 |
| | Mean | 0.6806 | 0.7479 | 0.7494 | 490.0 | **0.7522** | **0.7558** | 360.2 | 0.7420 | 0.7437 | 250.7 | **0.7542** | **0.7538** | 335.7 |
| Mean | | 0.6806 | 0.7105 | 0.7118 | 469.9 | **0.7171** | **0.7185** | 362.7 | 0.7078 | 0.7092 | 255.9 | **0.7187** | **0.7183** | 331.9 |

## 5.3 EFFECT OF SEMANTIC CONTEXT

To examine the impact of dataset semantic context, we compare the full version of FEBP with the semantically blinded version where the descriptions of datasets are removed (Appendix C.2). From Table 3, the full version outperforms the blinded version in terms of the mean performance score using all three downstream models. The Friedman-Nemenyi test shows that the performance difference is statistically significant at the $p = 0.01$ level. The performance difference is more pronounced for Random Forests and LightGBM, likely because the inclusion of non-semantically meaningful features consumes model capacity and causes greater overfitting to the training data.

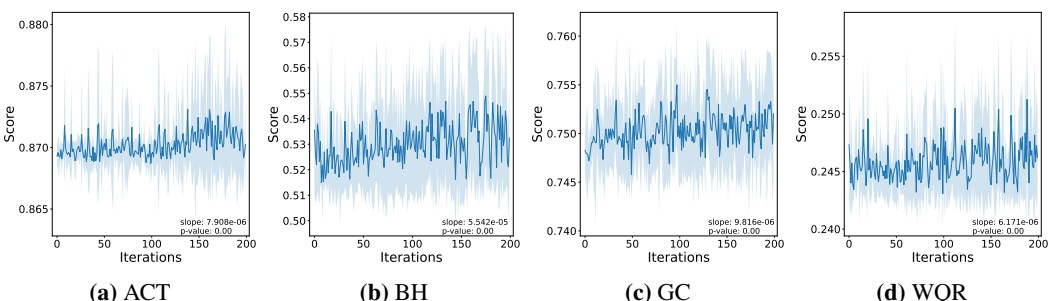

**Figure 5:** The cross validation score of candidate features on training data across iterations.

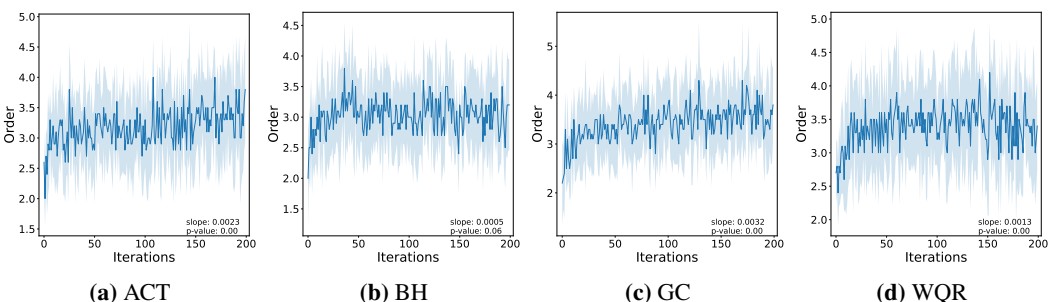

**Figure 6:** The order of candidate features across iterations.

We also report the number of LLM responses to help assess feature construction efficiency. As shown in Table 3, GPT-4 constructs features more efficiently than GPT-3.5 due to its broader knowledge. While incorporating dataset semantic information improves the feature construction efficiency of GPT-3.5, it reduces that of GPT-4. This is because the semantic information introduces bias, leading GPT-4 to generate more similar responses.

## 5.4 PERFORMANCE ANALYSIS

We analyze our LLM-based feature search process for deeper insights. Here, we present experimental results on the ACT, BH, GC, and WQR datasets using linear models from ten repeated runs with *gpt-3.5-turbo-0125*. The plots display the slope and $p$-value from one-tailed t-tests in OLS regressions, with the shaded area representing one standard deviation above and below the mean.

**Feature Learning.** We examine the cross validation score of candidate features on the training data across feature construction iterations. Figure 5 shows a general upward trend in the score as the number of iterations increases. This demonstrates that FEBP effectively improves the quality of constructed features through in-context learning of top-performing examples during feature search.

**Feature Complexity.** We analyze the order of candidate features across feature construction iterations. Figure 6 shows that the feature order increases more rapidly in the early iterations and stabilizes over time. On the one hand, FEBP effectively explores more complex features within promising feature spaces. On the other hand, our constraint instruction regularizes the process, preventing the generation of overly complex features.

**Feature Divergence.** We analyze the divergence of a new candidate features from previous ones during feature search. To measure this, we compute the edit distance between canonical feature expression trees using the algorithm from (Zhang & Shasha, 1989), normalizing the distance by the total number of nodes in both trees. Figure 7 displays the mean normalized tree edit distance between the current feature and the previous five features across iterations. The observed downward trend indicates that the feature search is converging.

**Feature Construction Efficiency.** We examine the number of LLM responses required to construct new candidate features. Figure 8 shows an upward trend over iterations, indicating that more responses are discarded. This is due to the increasing difficulty of constructing non-duplicate features

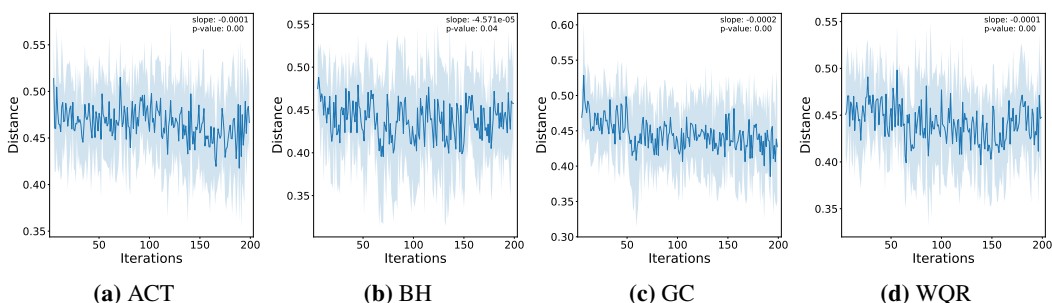

**Figure 7:** The mean normalized tree edit distance between a new candidate feature and previous five features.

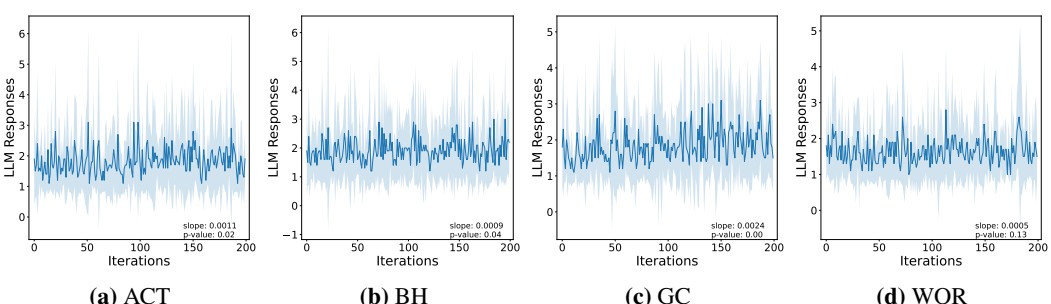

**Figure 8:** The number of LLM responses to construct a new candidate feature across iterations.

and the higher likelihood of syntactical errors as features become more complex. However, since the increase is small, FEBP remains scalable across a large number of iterations.

## 5.5 Hyperparameter Effect

**Number of Iterations.** Figure 9 shows the validation scores on the AF and CD datasets, which contain the smallest and largest numbers of features, respectively, using Random Forests and Light-GBM. The validation score is evaluated after adding the selected set of candidate features to the dataset, as indicated by $s_{n^*}$ in line 17 of Algorithm 1. We terminate our algorithm once we have 200 candidate features, as constructing additional features does not substantially enhance the performance, but constructing fewer features degrades the performance in some cases.

**Temperature.** Table 4 reports the maximum validation score across iterations along with the number of LLM responses, under different temperature settings. We observe that the best performance and feature construction efficiency are achieved when the temperature is set to 1. Lower temperatures increase the likelihood of the LLM repeating existing features, while higher temperatures make the LLM more prone to generating errors in responses, both reducing feature construction efficiency.

**Number of Examples in Prompt.** Table 5 reports the maximum validation score across iterations along with the number of LLM responses, using varying numbers of example features in the prompt.

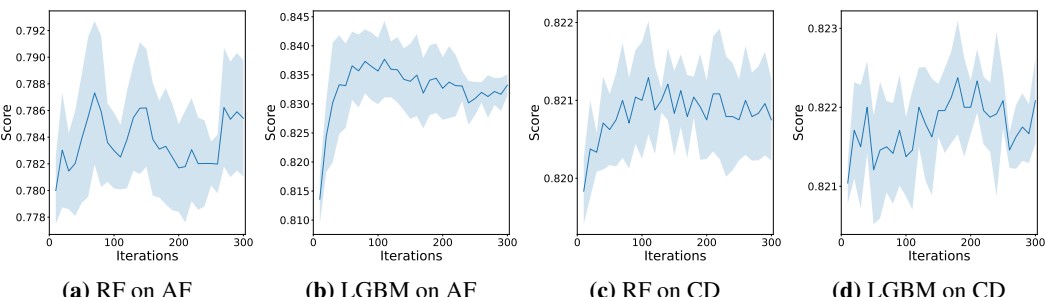

**Figure 9:** The validation score across iterations using Random Forests and LightGBM.

**Table 4:** Effect of the LLM temperature. For each compared setting, the left column shows the validation score, and the right column shows the number of LLM responses.

| Model | Dataset | Temperature | | | | | |
| | | 0.5 | | 1 | | 1.5 | |
|---|---|---|---|---|---|---|---|
| RF | AF | 0.7875 | 794.4 | 0.7914 | 393.2 | 0.7916 | 609.2 |
| | CD | 0.8211 | 823.2 | 0.8219 | 349.8 | 0.8218 | 672.6 |
| LGBM | AF | 0.8365 | 1313.2 | 0.8430 | 380.2 | 0.8418 | 627.6 |
| | CD | 0.8225 | 519.8 | 0.8226 | 352.2 | 0.8223 | 662.6 |
| Mean | | 0.8169 | 862.7 | 0.8197 | 368.9 | 0.8194 | 643.0 |

**Table 5:** Effect of the number of example features in the prompt. For each compared setting, the left column shows the validation score, and the right column shows the number of LLM responses.

| Model | Dataset | Number of Examples | | | | | | | |
| | | 1 | | 5 | | 10 | | 20 | |
|---|---|---|---|---|---|---|---|---|---|
| RF | AF | 0.7910 | 464.0 | 0.7930 | 409.2 | 0.7914 | 393.2 | 0.7860 | 372.8 |
| | WQR | 0.3897 | 339.4 | 0.3937 | 329.8 | 0.3948 | 362.6 | 0.3940 | 330.0 |
| | CD | 0.8215 | 429.6 | 0.8213 | 371.2 | 0.8219 | 349.8 | 0.8218 | 343.2 |
| LGBM | AF | 0.8421 | 440.4 | 0.8433 | 404.6 | 0.8430 | 380.2 | 0.8420 | 384.2 |
| | WQR | 0.4265 | 336.6 | 0.4294 | 334.8 | 0.4301 | 322.8 | 0.4333 | 330.4 |
| | CD | 0.8228 | 449.4 | 0.8224 | 361.2 | 0.8226 | 352.2 | 0.8228 | 321.2 |
| Mean | | 0.6823 | 409.9 | 0.6839 | 368.5 | 0.6840 | 360.1 | 0.6833 | 347.0 |

We observe that the best performance is achieved with 10 examples. Additionally, we observe that feature construction efficiency improves as the number of examples increases, as this helps the LLM reduce errors and generate more diverse responses. However, providing too many examples can hinder the in-context learning of optimal feature patterns, as shown by the performance decline.

# 6 CONCLUSION

We propose a novel LLM-based AutoFE algorithm for effective, efficient, and interpretable feature engineering that leverages the semantic information of datasets. Our approach provides the LLM with dataset descriptions and example features represented in canonical RPN, prompting it to construct new features. The LLM iteratively explores the feature space and improves feature construction by learning from top-performing examples. Experimental results demonstrate that our approach significantly outperforms state-of-the-art AutoFE methods and the inclusion of semantic context from dataset descriptions enhances performance. We also analyze the behavior of our LLM-based feature search process. Our work paves the way for further LLM-driven applications on automated machine learning pipelines and underscores the potential of utilizing semantic information. In the future, we plan to incorporate adaptive techniques for prompt design.

## ETHICAL STATEMENT

All datasets used in this work are publicly available, free of personal information, and intended for research purposes only. Our use of GPT models complies with the terms and conditions of OpenAI.

## REPRODUCIBILITY STATEMENT

The anonymized source code of this work can be accessed at `https://anonymous.4open.science/r/FEBP`.

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

# A  DISCUSSION ON CANONICAL RPN FEATURE REPRESENTATION

## A.1  WHY RPN

RPN provides a compact and unambiguous form of feature representation. In contrast, infix expression requires extra information such as brackets to determine operator precedence. Without brackets, the feature in infix expression col-0 − ( col-1 + col-2 ) would be indistinguishable from the feature ( col-0 − col-1 ) + col-2, while both features are distinctively encoded in RPN. Such compactness and unambiguity of RPN facilitate sequential modeling since there is no need to model the extra information, e.g., the positions of brackets.

Compared with other forms of feature representation such as prefix expression of depth-first traversal or breadth-first traversal, RPN better encodes the recursive structure of the expression tree. The bottom-up enumeration of tree nodes makes it easy for the LLM to evaluate the feature expression by scanning the sequence from left to right, for instance, ((col-0 col-1 −) col-2 +) (parentheses denote recursion). Using the prefix expression (+ (− col-0 col-1) col-2) or breadth-first expression (+ (− [col-2] col-0 col-1)), however, the LLM always needs to look back to find the operator, which undermines sequential modeling. We find in our experiments that when using prefix expression, the LLM encounters difficulty in generating syntactically valid feature expressions.

## A.2  WHY CANONICALIZATION

While there is one-to-one mapping between feature expression trees and RPN expressions, a feature that contains commutative operators (like addition and multiplication) can be represented by different RPN expressions, since the child nodes of these operators are unordered. We introduce a canonicalization scheme: arranging operator nodes before feature nodes and lexicographically sorting the nodes within each group. Through canonicalization, we create one-to-one mapping between features and cRPN expressions. This ensures the consistency of our feature representation and facilitates the in-context learning of feature patterns.

By arranging operator nodes before feature nodes, we also introduce left skewness to the expression tree that enhances the clarity of the recursive structure in cRPN. As illustrated in Figure 10, the original feature expression (col-2 (col-1 col-0 +) ∗) becomes ((col-0 col-1 +) col-2 ∗) after canonicalization, so that the LLM does not need to look back for col-2 when evaluating the expression. We present additional experimental results in Appendix E to validate the effectiveness of our canonicalization scheme.

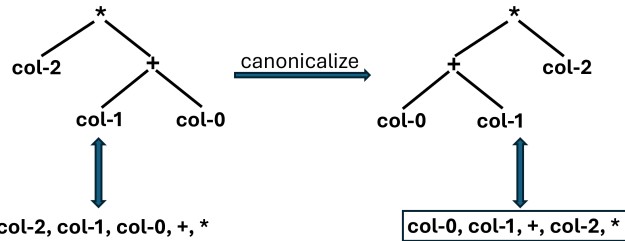

**Figure 10:** Our canonicalization scheme introduces left skewness to the expression tree.

# B  CONVERSION BETWEEN FEATURE EXPRESSION TREE AND RPN

Algorithms 2 and 3 detail the process of conversion between a feature expression tree and an RPN feature string. We check the RPN syntactical validity of a feature string in Algorithm 3 by checking whether there is enough child node in the stack in line 6 and the size of the stack is exactly one (the root) in line 13 returning the output.

---

**Algorithm 2:** Feature Expression Tree to RPN

---

**Input**  : A feature expression tree $T$
**Output:** An RPN feature string $f$

1  $r \leftarrow$ the root of $T$
2  Initialize string $f \leftarrow \epsilon$, stack $S \leftarrow [r]$, and $visited \leftarrow \emptyset$
3  **repeat**
4      $u \leftarrow S.peek()$
5      **if** $u \in visited$ **then**
6          $f.append(u)$
7          $S.pop()$
8      **end**
9      **else**
10         **for** each child $v$ of $u$ in the reverse order **do**
11             $S.push(v)$
12         **end**
13         $visited \leftarrow visited \cup \{u\}$
14     **end**
15 **until** $S$ is empty
16 **return** $f$

---

---

**Algorithm 3:** RPN to Feature Expression Tree

---

**Input**  : An RPN feature string $f$
**Output:** The root of a feature expression tree $T$

1  Initialize stack $S \leftarrow []$
2  **for** $i \leftarrow 1$ **to** $|f|$ **do**
3      $u \leftarrow$ the $i$-th element of $f$
4      **if** $u$ is an operator **then**
5          $o \leftarrow$ the arity of $u$
6          **for** $j \leftarrow 1$ **to** $o$ **do**
7              $v \leftarrow S.pop()$
8              Prepend $v$ to the list of children of $u$
9          **end**
10     **end**
11     $S.push(u)$
12 **end**
13 **return** $S.pop()$

---

## C EXAMPLE PROMPT

### C.1 FULL PROMPT

Figure 11 shows an example of full prompts used in our main experiments.

---

**Figure 11: Example full prompt on the Credit Default dataset.**

Dataset description:
This dataset contains information on default payments, demographic factors, credit data, history of payment, and bill statements of credit card clients in Taiwan from April 2005 to September 2005.
Dataset contains the following columns:
col-0 (int) [10000, 800000]: LIMIT_BAL: Amount of given credit in NT dollars (includes individual and family/supplementary credit
col-1 (category) {1, 2}: SEX: Gender (1=male, 2=female)
col-2 (category) {0, 1, 2, 3, 4, 5, 6}: EDUCATION: (1=graduate school, 2=university, 3=high school, 4=others, 5=unknown, 6=unknown)
col-3 (category) {0, 1, 2, 3}: MARRIAGE: Marital status (1=married, 2=single, 3=others)
col-4 (int) [21, 79]: AGE: Age in years
col-5 (category) {-2, -1, 0, 1, 2, 3, 4, 5, 6, 7, 8}: PAY_0: Repayment status in September, 2005 (-1=pay duly, 1=payment delay for one month, 2=payment delay for two months, ... 8=payment delay for eight months, 9=payment delay for nine months and above)
...
col-23 (category) {0, 1}: default.payment.next.month: Default payment (1=yes, 0=no)
We have the following unary operators:
log: taking the log of the absolute value
sqrt_abs: taking the square root of the absolute value
min_max: min-max normalization
reciprocal: taking the reciprocal
We have the following binary operators:
+: summing two columns
−: subtracting two columns
∗: multiplying two columns
/: taking the division of two columns
mod_column: taking the modulo of two columns
Feature strings are reverse Polish notation (RPN) expressions that operate on the columns of our dataset. Each feature string constructs an extra column that is useful for the downstream model Random Forests to predict the target col-23. The model will be trained on the dataset with the constructed columns and evaluated on a holdout set. The best columns will be selected.
Below are feature strings arranged in ascending order based on their performance scores. Higher scores are better.

Feature
col-17,col-21,*,col-20,+,sqrt_abs
Score
0.0011
...
Feature
col-4,col-6,*,col-12,col-16,-,sqrt_abs,*
Score
0.0014

Give me a new feature string that is different from all strings above and has a higher score. Use no more than five operators. Make sure all columns and operators exist and do not include the target column. Follow the syntax of RPN.

Output format:
Feature

(Feature name and description)

Usefulness
(Explanation why this adds useful real world knowledge to predict the target col-23 according to dataset description)

---

## C.2 SEMANTICALLY BLINDED PROMPT

Figure 12 shows an example of semantically blinded prompts used in our experiments in Section 5.3.

---

**Figure 12: Example semantically blinded prompt on the Credit Default dataset.**

Dataset contains the following columns:
col-0
col-1
col-2
col-3
col-4
col-5
. . .
col-23
We have the following unary operators:
log: taking the log of the absolute value
sqrt_abs: taking the square root of the absolute value
min_max: min-max normalization
reciprocal: taking the reciprocal
We have the following binary operators:
+: summing two columns
−: subtracting two columns
∗: multiplying two columns
/: taking the division of two columns
mod_column: taking the modulo of two columns
Feature strings are reverse Polish notation (RPN) expressions that operate on the columns of our dataset. Each feature string constructs an extra column that is useful for the downstream model Random Forests to predict the target col-23. The model will be trained on the dataset with the constructed columns and evaluated on a holdout set. The best columns will be selected.
Below are feature strings arranged in ascending order based on their performance scores. Higher scores are better.

Feature
col-17,col-21,*,col-20,+,sqrt_abs
Score
0.0011
. . .
Feature
col-4,col-6,*,col-12,col-16,-,sqrt_abs,*
Score
0.0014

Give me a new feature string that is different from all strings above and has a higher score. Use no more than five operators. Make sure all columns and operators exist and do not include the target column. Follow the syntax of RPN.

Output format:
Feature

(Feature name and description)

Usefulness
(Explanation why this adds useful real world knowledge to predict the target col-23 according to dataset description)

---

# D   EXPERIMENTAL DETAIL

## D.1   DATASET SOURCES

Table 6 summarizes the sources of datasets used in our experiments. Datasets are selected such that they cover different domains and both regression and classification tasks. Most of them have been used in previous works (Zhu et al., 2022a;b; Zhang et al., 2023; Hollmann et al., 2023).

**Table 6:** Sources of datasets.

| Name | Source |
|---|---|
| Airfoil (AF) | https://archive.ics.uci.edu/dataset/291/airfoil+self+noise |
| Boston Housing (BH) | https://www.kaggle.com/datasets/arunjangir245/boston-housing-dataset |
| Bikeshare (BS) | https://www.kaggle.com/datasets/marklvl/bike-sharing-dataset |
| Wine Quality Red (WQR) | https://archive.ics.uci.edu/dataset/186/wine+quality |
| AIDS Clinical Trials (ACT) | https://archive.ics.uci.edu/dataset/890/aids+clinical+trials+group+study+175 |
| Credit Default (CD) | https://www.kaggle.com/datasets/uciml/default-of-credit-card-clients-dataset |
| German Credit (GC) | https://archive.ics.uci.edu/dataset/573/south+german+credit+update |

## D.2   EXPERIMENTAL PLATFORM

All experiments are conducted on the Ubuntu 22.04.4 LTS operating system, 16 Intel(R) Core(TM) i7-7820X CPUs, and 4 NVIDIA GeForce RTX 2080 Ti GPUs, with the framework of Python 3.11.9 and PyTorch 1.12.1.

## D.3   FEATURE TRANSFORMATION OPERATORS

We list the details of all feature transformation operators below.

Unary transformations:

- Logarithm: Element-wise logarithm of the absolute value;
- Reciprocal: Element-wise reciprocal;
- Square root: Element-wise square root of the absolute value;
- Min-max normalization: Element-wise min-max normalization, with the min and max values from the training data.

Binary transformations:

- Addition: Element-wise addition;
- Subtraction: Element-wise subtraction;
- Multiplication: Element-wise multiplication;
- Division: Element-wise division;
- Modulo: Element-wise modulo.

### D.4 PARAMETER TUNING OF DOWNSTREAM MODELS

We tune the parameters of downstream models prior to and post AutoFE using randomized search implemented in an Sklearn package[3]. Table 7 lists the configurations of parameter tuning for each downstream model. We set the number of randomized search iterations to 100.

**Table 7:** Hyperparameter search space for downstream models.

| Model | Parameter | Search Space |
|---|---|---|
| Linear Model | regularization | loguniform(0.00001, 100) |
| Random Forests | num estimators | randint(5, 250) |
| | max depth | randint(1, 250) |
| | max features | uniform(0.01, 0.99) |
| | max samples | uniform(0.1, 0.9) |
| LightGBM | num estimators | randint(10, 1000) |
| | num leaves | randint(8, 64) |
| | learning rate | loguniform(0.001, 1) |
| | bagging fraction | uniform(0.1, 0.9) |
| | feature fraction | uniform(0.1, 0.9) |
| | reg lambda | loguniform(0.001, 100) |

### D.5 RELATIVE PERFORMANCE IMPROVEMENT

Tables 8 and 9 report the percentage improvement of FEBP over the baseline methods with GPT-3.5 and GPT-4, respectively, corresponding to the experimental results in Table 2.

### D.6 STANDARD DEVIATIONS

Tables 10-13 report the sample standard deviations corresponding to the experimental results in Tables 2-5, respectively.

### D.7 STATISTICAL TESTS

We perform the Friedman test (Friedman, 1937) to determine whether there is statistically significant difference among the compared AutoFE methods. The Friedman test $p$-values for the results in Tables 2 and 3 are $1.16 \times 10^{-47}$ and $3.95 \times 10^{-34}$, respectively. Hence, we can reject the null hypothesis that the performance is the same for all methods. We perform the Nemenyi post-hoc test (Nemenyi, 1963) to further determine which AutoFE methods have different performance. Tables 14 and 15 summarize the $p$-values for the pairwise comparisons in Tables 2 and 3, respectively. From Table 14, the performance difference between our method FEBP and baseline methods other than DIFER (Zhu et al., 2022b) is statistically significant at the $p = 0.01$ level. From Table 15, the performance difference between the full version of FEBP and the semantically blinded version is statistically significant at the $p = 0.01$ level.

To highlight the performance difference when using Random Forests and LightGBM, we perform additional statistical tests for the results in Table 2 excluding the linear model results. The Friedman test $p$-value is $6.14 \times 10^{-23}$. Table 16 summarizes the $p$-values from the Nemenyi post-hoc test for pairwise comparison. We observe that FEBP with GPT-3.5 and post-AutoFE parameter tuning significantly outperforms all baselines except DIFER at the $p = 0.05$ level. With GPT-4, the performance difference between FEBP and CAAFE (Hollmann et al., 2023) is statistically significant at the $p = 0.05$ level.

---

[3]`https://scikit-learn.org/1.5/modules/generated/sklearn.model_selection.RandomizedSearchCV.html`

**Table 8:** The percentage improvement of FEBP over the baseline methods, with GPT-3.5. For each compared method, the left and right columns show the results without and with parameter tuning of the downstream model algorithm post AutoFE, respectively.

| Model | Dataset | Raw | | DIFER | | OpenFE | | CAAFE | |
|---|---|---|---|---|---|---|---|---|---|
| | AF | 90.34 | 90.46 | 12.65 | 8.64 | 53.77 | 53.77 | 64.86 | 64.76 |
| | BH | 32.27 | 33.06 | -0.37 | 0.61 | 28.06 | 29.51 | 4.32 | 5.46 |
| Linear | WQR | 0.96 | 1.80 | 9.97 | 4.37 | 0.35 | 0.32 | -0.74 | -0.46 |
| Model | ACT | 2.64 | 3.41 | 0.16 | -0.05 | 0.00 | 0.75 | 2.47 | 3.29 |
| | CD | 0.18 | 0.19 | 0.10 | 0.03 | 0.21 | 0.17 | 0.20 | 0.19 |
| | GC | 6.62 | 5.07 | 6.02 | 0.54 | 3.42 | 2.47 | 2.99 | 1.77 |
| Mean | | 22.17 | 22.33 | 4.76 | 2.36 | 14.30 | 14.50 | 12.35 | 12.50 |
| | AF | 0.42 | 1.44 | 0.78 | 0.02 | 1.72 | 1.37 | -0.02 | 1.23 |
| | BH | 2.26 | 1.97 | -2.95 | -2.95 | -1.92 | -1.55 | -0.13 | -0.41 |
| Random | BS | 4.52 | 4.60 | 0.08 | 0.10 | -0.29 | -0.21 | -0.43 | -0.35 |
| Forests | WQR | 5.44 | 5.00 | 0.61 | 0.34 | 2.89 | 3.11 | 3.86 | 3.42 |
| | ACT | 1.33 | 1.27 | 0.32 | 0.26 | 1.06 | 0.90 | 1.11 | 0.74 |
| | CD | 0.02 | 0.01 | 0.12 | 0.04 | 0.09 | 0.10 | 0.05 | 0.06 |
| | GC | 2.55 | 2.28 | 1.19 | 1.60 | -0.13 | 0.66 | -0.65 | 0.00 |
| Mean | | 2.36 | 2.37 | 0.02 | -0.08 | 0.49 | 0.63 | 0.54 | 0.67 |
| | AF | -0.76 | 0.20 | 0.32 | -0.23 | 1.51 | 1.80 | -0.63 | 0.53 |
| | BH | 1.48 | 1.94 | 0.21 | 0.14 | -1.30 | 0.47 | 1.42 | 1.32 |
| Light- | BS | 3.27 | 3.45 | -0.27 | -0.32 | -0.15 | -0.43 | 1.91 | 1.98 |
| GBM | WQR | 7.67 | 9.04 | -0.63 | -0.27 | 5.66 | 7.40 | -0.29 | 3.36 |
| | ACT | 0.63 | 1.06 | 1.06 | 1.11 | 0.90 | 1.43 | 0.74 | 0.74 |
| | CD | 0.02 | -0.04 | 0.22 | 0.25 | 0.10 | 0.12 | 0.06 | -0.01 |
| | GC | 5.93 | 6.48 | 0.39 | 1.58 | 1.72 | 0.26 | 2.54 | 2.25 |
| Mean | | 2.60 | 3.16 | 0.18 | 0.32 | 1.21 | 1.58 | 0.82 | 1.45 |
| Mean | | 8.39 | 8.63 | 1.50 | 0.79 | 4.88 | 5.12 | 4.18 | 4.49 |

**Table 9:** The percentage improvement of FEBP over the baseline methods, with GPT-4. For each compared method, the left and right columns show the results without and with parameter tuning of the downstream model algorithm post AutoFE, respectively.

| Model | Dataset | Raw | | DIFER | | OpenFE | | CAAFE | |
|---|---|---|---|---|---|---|---|---|---|
| | AF | 91.40 | 91.34 | 13.27 | 9.14 | 54.62 | 54.48 | 51.94 | 51.82 |
| | BH | 37.28 | 40.06 | 3.41 | 5.90 | 32.92 | 36.33 | 15.14 | 17.39 |
| Linear | WQR | 0.64 | 1.94 | 9.62 | 4.51 | 0.03 | 0.46 | -2.25 | -0.99 |
| Model | ACT | 3.08 | 3.02 | 0.59 | -0.42 | 0.43 | 0.37 | 2.35 | 2.24 |
| | CD | 0.26 | 0.26 | 0.18 | 0.10 | 0.28 | 0.25 | 0.61 | 0.61 |
| | GC | 6.90 | 4.51 | 6.30 | 0.00 | 3.69 | 1.92 | 5.27 | 2.91 |
| Mean | | 23.26 | 23.52 | 5.56 | 3.21 | 15.33 | 15.64 | 12.17 | 12.33 |
| | AF | 0.05 | 0.93 | 0.41 | -0.47 | 1.35 | 0.86 | -0.20 | 0.37 |
| | BH | 2.16 | 1.76 | -3.05 | -3.15 | -2.02 | -1.76 | 0.56 | 0.54 |
| Random | BS | 4.23 | 4.25 | -0.20 | -0.23 | -0.57 | -0.54 | 0.28 | 0.32 |
| Forests | WQR | 4.03 | 4.03 | -0.74 | -0.58 | 1.51 | 2.16 | 3.17 | 3.17 |
| | ACT | 0.95 | 0.64 | -0.05 | -0.37 | 0.69 | 0.26 | 0.74 | 0.42 |
| | CD | 0.02 | -0.20 | 0.13 | -0.18 | 0.10 | -0.11 | 0.02 | -0.13 |
| | GC | 3.09 | 3.09 | 1.72 | 2.40 | 0.39 | 1.45 | 0.26 | 0.66 |
| Mean | | 2.08 | 2.07 | -0.26 | -0.37 | 0.21 | 0.33 | 0.69 | 0.76 |
| | AF | -0.11 | 0.24 | 0.98 | -0.19 | 2.18 | 1.84 | -0.75 | -0.36 |
| | BH | 1.90 | 1.04 | 0.63 | -0.74 | -0.89 | -0.41 | 3.00 | 1.70 |
| Light- | BS | 3.94 | 4.08 | 0.38 | 0.28 | 0.51 | 0.17 | 3.72 | 3.44 |
| GBM | WQR | 5.12 | 5.67 | -2.98 | -3.35 | 3.16 | 4.08 | 3.04 | 2.28 |
| | ACT | 0.79 | 1.06 | 1.22 | 1.11 | 1.06 | 1.43 | 0.85 | 1.22 |
| | CD | 0.04 | -0.07 | 0.24 | 0.21 | 0.12 | 0.08 | 0.03 | 0.00 |
| | GC | 7.03 | 6.21 | 1.44 | 1.32 | 2.78 | 0.00 | 4.16 | -0.26 |
| Mean | | 2.67 | 2.60 | 0.27 | -0.19 | 1.27 | 1.03 | 2.01 | 1.15 |
| Mean | | 8.64 | 8.69 | 1.67 | 0.76 | 5.12 | 5.17 | 4.60 | 4.37 |

**Table 10:** Standard deviations of Table 2, summary of experimental results.

| Model | Dataset | Raw | DIFER | | OpenFE | | CAAFE GPT-3.5 | | GPT-4 | | FEBP (ours) GPT-3.5 | | GPT-4 | |
|---|---|---|---|---|---|---|---|---|---|---|---|---|---|---|
| Linear Model | AF | – | 0.2559 | 0.2012 | 0.0015 | 0.0014 | 0.0099 | 0.0102 | 0.0511 | 0.0513 | 0.0101 | 0.0100 | 0.0267 | 0.0268 |
| | BH | – | 0.0092 | 0.0153 | 0.0169 | 0.0188 | 0.0196 | 0.0184 | 0.0408 | 0.0419 | 0.0111 | 0.0149 | 0.0254 | 0.0184 |
| | WQR | – | 0.0305 | 0.0223 | 0.0058 | 0.0055 | 0.0046 | 0.0038 | 0.0060 | 0.0060 | 0.0135 | 0.0112 | 0.0068 | 0.0044 |
| | ACT | – | 0.0179 | 0.0073 | 0.0140 | 0.0105 | 0.0035 | 0.0021 | 0.0054 | 0.0053 | 0.0085 | 0.0051 | 0.0040 | 0.0062 |
| | CD | – | 0.0014 | 0.0006 | 0.0006 | 0.0002 | 0.0006 | 0.0007 | 0.0057 | 0.0051 | 0.0013 | 0.0007 | 0.0006 | 0.0009 |
| | GC | – | 0.0272 | 0.0104 | 0.0097 | 0.0076 | 0.0100 | 0.0125 | 0.0134 | 0.0108 | 0.0120 | 0.0213 | 0.0108 | 0.0152 |
| Random Forests | AF | – | 0.0054 | 0.0044 | 0.0032 | 0.0036 | 0.0032 | 0.0034 | 0.0108 | 0.0084 | 0.0090 | 0.0086 | 0.0059 | 0.0095 |
| | BH | – | 0.0142 | 0.0131 | 0.0034 | 0.0068 | 0.0050 | 0.0050 | 0.0084 | 0.0113 | 0.0057 | 0.0077 | 0.0059 | 0.0046 |
| | BS | – | 0.0128 | 0.0113 | 0.0003 | 0.0003 | 0.0003 | 0.0003 | 0.0208 | 0.0207 | 0.0088 | 0.0070 | 0.0157 | 0.0154 |
| | WQR | – | 0.0108 | 0.0109 | 0.0030 | 0.0076 | 0.0022 | 0.0022 | 0.0051 | 0.0051 | 0.0034 | 0.0069 | 0.0022 | 0.0026 |
| | ACT | – | 0.0048 | 0.0058 | 0.0037 | 0.0087 | 0.0030 | 0.0055 | 0.0020 | 0.0030 | 0.0055 | 0.0051 | 0.0043 | 0.0054 |
| | CD | – | 0.0010 | 0.0011 | 0.0003 | 0.0004 | 0.0005 | 0.0004 | 0.0008 | 0.0001 | 0.0011 | 0.0010 | 0.0009 | 0.0017 |
| | GC | – | 0.0184 | 0.0177 | 0.0154 | 0.0110 | 0.0082 | 0.0076 | 0.0065 | 0.0164 | 0.0114 | 0.0067 | 0.0097 | 0.0097 |
| Light-GBM | AF | – | 0.0029 | 0.0029 | 0.0058 | 0.0036 | 0.0067 | 0.0027 | 0.0072 | 0.0077 | 0.0129 | 0.0054 | 0.0061 | 0.0041 |
| | BH | – | 0.0147 | 0.0260 | 0.0128 | 0.0150 | 0.0114 | 0.0111 | 0.0145 | 0.0188 | 0.0169 | 0.0076 | 0.0134 | 0.0073 |
| | BS | – | 0.0092 | 0.0070 | 0.0007 | 0.0004 | 0.0159 | 0.0198 | 0.0056 | 0.0139 | 0.0151 | 0.0139 | 0.0033 | 0.0034 |
| | WQR | – | 0.0134 | 0.0164 | 0.0072 | 0.0133 | 0.0084 | 0.0080 | 0.0116 | 0.0134 | 0.0123 | 0.0085 | 0.0097 | 0.0092 |
| | ACT | – | 0.0048 | 0.0042 | 0.0068 | 0.0094 | 0.0061 | 0.0045 | 0.0045 | 0.0027 | 0.0027 | 0.0017 | 0.0050 | 0.0077 |
| | CD | – | 0.0009 | 0.0013 | 0.0004 | 0.0010 | 0.0008 | 0.0005 | 0.0010 | 0.0007 | 0.0004 | 0.0004 | 0.0004 | 0.0008 |
| | GC | – | 0.0141 | 0.0184 | 0.0184 | 0.0184 | 0.0222 | 0.0166 | 0.0079 | 0.0199 | 0.0076 | 0.0045 | 0.0096 | 0.0146 |

**Table 11:** Standard deviations of Table 3, performance comparison of FEBP with and without semantic blinding.

| Model | Dataset | Raw | GPT-3.5 Blinded | | | Full | | | GPT-4 Blinded | | | Full | | |
|---|---|---|---|---|---|---|---|---|---|---|---|---|---|---|
| Linear Model | AF | – | 0.0147 | 0.0156 | 36.1 | 0.0101 | 0.0100 | 28.8 | 0.0162 | 0.0161 | 25.8 | 0.0267 | 0.0268 | 92.3 |
| | BH | – | 0.0444 | 0.0519 | 39.0 | 0.0111 | 0.0149 | 42.2 | 0.0161 | 0.0131 | 66.7 | 0.0254 | 0.0184 | 58.6 |
| | WQR | – | 0.0133 | 0.0032 | 48.9 | 0.0135 | 0.0112 | 15.3 | 0.0128 | 0.0046 | 23.5 | 0.0068 | 0.0044 | 80.6 |
| | ACT | – | 0.0088 | 0.0107 | 15.4 | 0.0085 | 0.0051 | 17.5 | 0.0056 | 0.0085 | 15.5 | 0.0040 | 0.0062 | 54.8 |
| | CD | – | 0.0014 | 0.0003 | 27.6 | 0.0013 | 0.0007 | 13.1 | 0.0021 | 0.0011 | 13.2 | 0.0006 | 0.0009 | 14.8 |
| | GC | – | 0.0114 | 0.0042 | 32.3 | 0.0120 | 0.0213 | 14.3 | 0.0125 | 0.0114 | 11.0 | 0.0108 | 0.0152 | 36.4 |
| Random Forests | AF | – | 0.0086 | 0.0058 | 60.3 | 0.0090 | 0.0086 | 47.3 | 0.0092 | 0.0079 | 27.9 | 0.0059 | 0.0095 | 93.6 |
| | BH | – | 0.0068 | 0.0068 | 45.3 | 0.0057 | 0.0077 | 14.5 | 0.0142 | 0.0132 | 24.7 | 0.0059 | 0.0046 | 23.0 |
| | BS | – | 0.0186 | 0.0181 | 112.1 | 0.0088 | 0.0070 | 47.8 | 0.0103 | 0.0088 | 38.8 | 0.0157 | 0.0154 | 39.2 |
| | WQR | – | 0.0078 | 0.0081 | 40.5 | 0.0034 | 0.0069 | 18.5 | 0.0092 | 0.0075 | 19.1 | 0.0022 | 0.0026 | 45.2 |
| | ACT | – | 0.0099 | 0.0035 | 33.7 | 0.0055 | 0.0051 | 13.1 | 0.0100 | 0.0093 | 16.6 | 0.0043 | 0.0054 | 85.7 |
| | CD | – | 0.0015 | 0.0008 | 53.3 | 0.0011 | 0.0010 | 14.5 | 0.0005 | 0.0008 | 83.4 | 0.0009 | 0.0017 | 56.9 |
| | GC | – | 0.0067 | 0.0057 | 28.9 | 0.0114 | 0.0067 | 17.3 | 0.0210 | 0.0143 | 12.8 | 0.0097 | 0.0097 | 113.1 |
| Light-GBM | AF | – | 0.0104 | 0.0060 | 66.8 | 0.0129 | 0.0054 | 21.7 | 0.0142 | 0.0155 | 39.6 | 0.0061 | 0.0041 | 73.1 |
| | BH | – | 0.0131 | 0.0170 | 60.7 | 0.0169 | 0.0076 | 20.7 | 0.0119 | 0.0121 | 25.7 | 0.0134 | 0.0073 | 36.1 |
| | BS | – | 0.0152 | 0.0178 | 76.3 | 0.0151 | 0.0139 | 31.8 | 0.0048 | 0.0049 | 74.5 | 0.0033 | 0.0034 | 32.1 |
| | WQR | – | 0.0151 | 0.0028 | 36.9 | 0.0123 | 0.0085 | 17.3 | 0.0195 | 0.0190 | 21.1 | 0.0097 | 0.0092 | 46.3 |
| | ACT | – | 0.0021 | 0.0030 | 44.2 | 0.0027 | 0.0017 | 28.5 | 0.0042 | 0.0128 | 15.7 | 0.0050 | 0.0077 | 49.6 |
| | CD | – | 0.0011 | 0.0011 | 59.4 | 0.0004 | 0.0004 | 15.7 | 0.0007 | 0.0010 | 5.6 | 0.0004 | 0.0008 | 85.7 |
| | GC | – | 0.0130 | 0.0148 | 41.7 | 0.0076 | 0.0045 | 23.0 | 0.0117 | 0.0094 | 13.7 | 0.0096 | 0.0146 | 46.9 |

**Table 12:** Standard deviations of Table 4, effect of temperature.

| Model | Dataset | Temperature 0.5 | | 1 | | 1.5 | |
|---|---|---|---|---|---|---|---|
| RF | AF | 0.0071 | 160.9 | 0.0042 | 47.3 | 0.0040 | 34.7 |
| | CD | 0.0005 | 324.3 | 0.0004 | 14.5 | 0.0005 | 64.1 |
| LGBM | AF | 0.0042 | 523.3 | 0.0044 | 21.7 | 0.0022 | 59.8 |
| | CD | 0.0008 | 174.7 | 0.0007 | 15.7 | 0.0005 | 73.0 |

**Table 13:** Standard deviations of Table 5, effect of the number of example features in the prompt.

| Model | Dataset | Number of Examples 1 | | 5 | | 10 | | 20 | |
|---|---|---|---|---|---|---|---|---|---|
| RF | AF | 0.0054 | 55.8 | 0.0035 | 45.0 | 0.0042 | 47.3 | 0.0056 | 24.0 |
| | WQR | 0.0088 | 19.6 | 0.0038 | 11.4 | 0.0027 | 18.5 | 0.0096 | 29.6 |
| | CD | 0.0005 | 46.5 | 0.0007 | 19.1 | 0.0004 | 14.5 | 0.0006 | 17.8 |
| LGBM | AF | 0.0065 | 103.2 | 0.0031 | 21.6 | 0.0044 | 21.7 | 0.0044 | 56.4 |
| | WQR | 0.0048 | 16.9 | 0.0057 | 32.4 | 0.0064 | 17.3 | 0.0064 | 26.5 |
| | CD | 0.0003 | 71.2 | 0.0002 | 39.0 | 0.0007 | 15.7 | 0.0005 | 17.5 |

**Table 14:** The Nemenyi post-hoc test $p$-values for pairwise comparison of the methods in Table 2. Results that are significant at the $p = 0.05$ confidence level are highlighted in boldface.

| | | Raw | DIFER | | OpenFE | | CAAFE GPT-3.5 | | CAAFE GPT-4 | | FEBP (ours) GPT-3.5 | | FEBP (ours) GPT-4 | |
|---|---|---|---|---|---|---|---|---|---|---|---|---|---|---|
| Raw | | 1.0000 | **0.0010** | **0.0010** | **0.0203** | **0.0086** | **0.0010** | **0.0010** | **0.0409** | **0.0179** | **0.0010** | **0.0010** | **0.0010** | **0.0010** |
| DIFER | | **0.0010** | 1.0000 | 0.3235 | 0.4051 | 0.5626 | 0.9000 | 0.9000 | 0.2697 | 0.4310 | **0.0397** | **0.0010** | **0.0028** | 0.1535 |
| | | **0.0010** | 0.3235 | 1.0000 | **0.0010** | **0.0010** | **0.0343** | **0.0397** | **0.0010** | **0.0010** | 0.9000 | 0.7526 | 0.9000 | 0.9000 |
| OpenFE | | **0.0203** | 0.4051 | **0.0010** | 1.0000 | 0.9000 | 0.9000 | 0.9000 | 0.9000 | 0.9000 | **0.0010** | **0.0010** | **0.0010** | **0.0010** |
| | | **0.0086** | 0.5626 | **0.0010** | 0.9000 | 1.0000 | 0.9000 | 0.9000 | 0.9000 | 0.9000 | **0.0010** | **0.0010** | **0.0010** | **0.0010** |
| CAAFE | GPT-3.5 | **0.0010** | 0.9000 | **0.0343** | 0.9000 | 0.9000 | 1.0000 | 0.9000 | 0.8216 | 0.9000 | **0.0016** | **0.0010** | **0.0010** | **0.0105** |
| | | **0.0010** | 0.9000 | **0.0397** | 0.9000 | 0.9000 | 0.9000 | 1.0000 | 0.7929 | 0.9000 | **0.0019** | **0.0010** | **0.0010** | **0.0125** |
| | GPT-4 | **0.0409** | 0.2697 | **0.0010** | 0.9000 | 0.9000 | 0.8216 | 0.7929 | 1.0000 | 0.9000 | **0.0010** | **0.0010** | **0.0010** | **0.0010** |
| | | **0.0179** | 0.4310 | **0.0010** | 0.9000 | 0.9000 | 0.9000 | 0.9000 | 0.9000 | 1.0000 | **0.0010** | **0.0010** | **0.0010** | **0.0010** |
| FEBP | GPT-3.5 | **0.0010** | **0.0397** | 0.9000 | **0.0010** | **0.0010** | **0.0016** | **0.0019** | **0.0010** | **0.0010** | 1.0000 | 0.9000 | 0.9000 | 0.9000 |
| | | **0.0010** | **0.0010** | 0.7526 | **0.0010** | **0.0010** | **0.0010** | **0.0010** | **0.0010** | **0.0010** | 0.9000 | 1.0000 | 0.9000 | 0.9000 |
| | GPT-4 | **0.0010** | **0.0028** | 0.9000 | **0.0010** | **0.0010** | **0.0010** | **0.0010** | **0.0010** | **0.0010** | 0.9000 | 0.9000 | 1.0000 | 0.9000 |
| | | **0.0010** | 0.1535 | 0.9000 | **0.0010** | **0.0010** | **0.0105** | **0.0125** | **0.0010** | **0.0010** | 0.9000 | 0.9000 | 0.9000 | 1.0000 |

**Table 15:** The Nemenyi post-hoc test $p$-values for pairwise comparison of the methods in Table 3. Results that are significant at the $p = 0.05$ confidence level are highlighted in boldface.

| | | Raw | GPT-3.5 Blinded | | GPT-3.5 Full | | GPT-4 Blinded | | GPT-4 Full | |
|---|---|---|---|---|---|---|---|---|---|---|
| Raw | | 1.0000 | **0.0010** | **0.0010** | **0.0010** | **0.0010** | **0.0017** | **0.0010** | **0.0010** | **0.0010** |
| GPT-3.5 | Blinded | **0.0010** | 1.0000 | 0.9000 | **0.0062** | **0.0010** | 0.9000 | 0.9000 | **0.0010** | **0.0057** |
| | | **0.0010** | 0.9000 | 1.0000 | 0.1775 | **0.0066** | 0.3858 | 0.9000 | **0.0105** | 0.1677 |
| | Full | **0.0010** | **0.0062** | 0.1775 | 1.0000 | 0.9000 | **0.0010** | **0.0069** | 0.9000 | 0.9000 |
| | | **0.0010** | **0.0010** | **0.0066** | 0.9000 | 1.0000 | **0.0010** | **0.0010** | 0.9000 | 0.9000 |
| GPT-4 | Blinded | **0.0017** | 0.9000 | 0.3858 | **0.0010** | **0.0010** | 1.0000 | 0.9000 | **0.0010** | **0.0010** |
| | | **0.0010** | 0.9000 | 0.9000 | **0.0069** | **0.0010** | 0.9000 | 1.0000 | **0.0010** | **0.0062** |
| | Full | **0.0010** | **0.0010** | **0.0105** | 0.9000 | 0.9000 | **0.0010** | **0.0010** | 1.0000 | 0.9000 |
| | | **0.0010** | **0.0057** | 0.1677 | 0.9000 | 0.9000 | **0.0010** | **0.0062** | 0.9000 | 1.0000 |

**Table 16:** The Nemenyi post-hoc test $p$-values for pairwise comparison of the methods in Table 2, excluding linear model results. Results that are significant at the $p = 0.05$ confidence level are highlighted in boldface.

| | | Raw | DIFER | | OpenFE | | CAAFE GPT-3.5 | | CAAFE GPT-4 | | FEBP (ours) GPT-3.5 | | FEBP (ours) GPT-4 | |
|---|---|---|---|---|---|---|---|---|---|---|---|---|---|---|
| Raw | | 1.0000 | **0.0010** | **0.0010** | 0.5006 | 0.3953 | **0.0010** | **0.0012** | 0.3875 | 0.2344 | **0.0010** | **0.0010** | **0.0010** | **0.0010** |
| DIFER | | **0.0010** | 1.0000 | 0.9000 | 0.6382 | 0.7345 | 0.9000 | 0.9000 | 0.7414 | 0.8996 | 0.4263 | **0.0299** | 0.1392 | 0.9000 |
| | | **0.0010** | 0.9000 | 1.0000 | **0.0098** | **0.0171** | 0.9000 | 0.8308 | **0.0178** | **0.0412** | 0.9000 | 0.8377 | 0.9000 | 0.9000 |
| OpenFE | | 0.5006 | 0.6382 | **0.0098** | 1.0000 | 0.9000 | 0.6175 | 0.7138 | 0.9000 | 0.9000 | **0.0010** | **0.0010** | **0.0010** | **0.0171** |
| | | 0.3953 | 0.7345 | **0.0171** | 0.9000 | 1.0000 | 0.7138 | 0.8102 | 0.9000 | 0.9000 | **0.0010** | **0.0010** | **0.0010** | **0.0289** |
| CAAFE | GPT-3.5 | **0.0010** | 0.9000 | 0.9000 | 0.6175 | 0.7138 | 1.0000 | 0.9000 | 0.7207 | 0.8790 | 0.4493 | **0.0334** | 0.1516 | 0.9000 |
| | | **0.0012** | 0.9000 | 0.8308 | 0.7138 | 0.8102 | 0.9000 | 1.0000 | 0.8170 | 0.9000 | 0.3422 | **0.0199** | 0.1017 | 0.9000 |
| | GPT-4 | 0.3875 | 0.7414 | **0.0178** | 0.9000 | 0.9000 | 0.7207 | 0.8170 | 1.0000 | 0.9000 | **0.0010** | **0.0010** | **0.0010** | **0.0299** |
| | | 0.2344 | 0.8996 | **0.0412** | 0.9000 | 0.9000 | 0.8790 | 0.9000 | 0.9000 | 1.0000 | **0.0026** | **0.0010** | **0.0010** | 0.0661 |
| FEBP | GPT-3.5 | **0.0010** | 0.4263 | 0.9000 | **0.0010** | **0.0010** | 0.4493 | 0.3422 | **0.0010** | **0.0026** | 1.0000 | 0.9000 | 0.9000 | 0.9000 |
| | | **0.0010** | **0.0299** | 0.8377 | **0.0010** | **0.0010** | **0.0334** | **0.0199** | **0.0010** | **0.0010** | 0.9000 | 1.0000 | 0.9000 | 0.7414 |
| | GPT-4 | **0.0010** | 0.1392 | 0.9000 | **0.0010** | **0.0010** | 0.1516 | 0.1017 | **0.0010** | **0.0010** | 0.9000 | 0.9000 | 1.0000 | 0.9000 |
| | | **0.0010** | 0.9000 | 0.9000 | **0.0171** | **0.0289** | 0.9000 | 0.9000 | **0.0299** | 0.0661 | 0.9000 | 0.7414 | 0.9000 | 1.0000 |

## D.8 NUMBER OF SELECTED FEATURES

Table 17 compares the number of features added to the datasets. Our method FEBP adaptively determines the number of features and selects fewer features than DIFER (Zhu et al., 2022b), demonstrating the effectiveness of the features generated by our method.

Table 17: Comparison of the number of selected features.

| Model | Dataset | DIFER | OpenFE | FEBP Blinded | | FEBP | |
|---|---|---|---|---|---|---|---|
| | | | | GPT-3.5 | GPT-4 | GPT-3.5 | GPT-4 |
| Linear Model | AF | 310 | 10 | 167 | 165 | 162 | 183 |
| | BH | 156 | 10 | 104 | 141 | 144 | 90 |
| | WQR | 109 | 10 | 57 | 80 | 43 | 55 |
| | ACT | 113 | 10 | 84 | 49 | 85 | 14 |
| | CD | 157 | 10 | 92 | 68 | 74 | 74 |
| | GC | 105 | 10 | 75 | 97 | 120 | 51 |
| Random Forests | AF | 387 | 10 | 39 | 19 | 15 | 34 |
| | BH | 186 | 10 | 4 | 6 | 19 | 77 |
| | BS | 46 | 10 | 9 | 7 | 9 | 65 |
| | WQR | 63 | 10 | 9 | 44 | 39 | 45 |
| | ACT | 339 | 10 | 55 | 35 | 69 | 61 |
| | CD | 178 | 10 | 97 | 74 | 94 | 89 |
| | GC | 92 | 10 | 68 | 84 | 31 | 59 |
| Light-GBM | AF | 325 | 10 | 30 | 55 | 42 | 24 |
| | BH | 118 | 10 | 15 | 17 | 16 | 25 |
| | BS | 287 | 10 | 119 | 48 | 68 | 116 |
| | WQR | 454 | 10 | 64 | 29 | 129 | 128 |
| | ACT | 132 | 10 | 54 | 46 | 16 | 51 |
| | CD | 409 | 10 | 68 | 53 | 12 | 50 |
| | GC | 501 | 10 | 61 | 86 | 16 | 35 |
| Mean | | 223 | 10 | 64 | 60 | 60 | 66 |

### D.9 COMPUTATION COST

Table 18 compares the number of features evaluated during the feature search process. Guided by domain knowledge, our method FEBP evaluates much fewer features than DIFER (Zhu et al., 2022b) and OpenFE (Zhang et al., 2023).

Tables 19 and 20 summarize the computation time, with *gpt-3.5-turbo-0125* as the LLM. For FEBP, the computation time of LLM generation and feature evaluation is relatively stable across datasets of varying sizes. We note that the LLM generation time can be substantially reduced by instructing the LLM to generate multiple features in a generation step.

**Table 18:** Comparison of the number of evaluated features during feature search.

| Model | Dataset | DIFER | OpenFE | FEBP |
|---|---|---|---|---|
| Linear Model | AF | 2083 | 224 | 200 |
| | BH | 2081 | 1167 | 200 |
| | WQR | 2083 | 929 | 200 |
| | ACT | 2077 | 4310 | 200 |
| | CD | 2088 | 3385 | 200 |
| | GC | 2076 | 4169 | 200 |
| Random Forests | AF | 2085 | 224 | 200 |
| | BH | 2079 | 1051 | 200 |
| | BS | 2082 | 310 | 200 |
| | WQR | 2085 | 929 | 200 |
| | ACT | 2079 | 1636 | 200 |
| | CD | 2086 | 1801 | 200 |
| | GC | 2078 | 2139 | 200 |
| Light-GBM | AF | 2084 | 224 | 200 |
| | BH | 2080 | 1051 | 200 |
| | BS | 2083 | 310 | 200 |
| | WQR | 2084 | 929 | 200 |
| | ACT | 2079 | 1636 | 200 |
| | CD | 2087 | 1801 | 200 |
| | GC | 2078 | 2139 | 200 |
| Mean | | 2082 | 1518 | 200 |

Table 19: Comparison of computation time, in minutes.

| Model | Dataset | DIFER | OpenFE | CAAFE | FEBP |
|---|---|---|---|---|---|
| Linear Model | AF | 33.49 | 0.21 | 1.73 | 42.80 |
| | BH | 41.17 | 0.21 | 1.18 | 41.28 |
| | WQR | 34.94 | 0.25 | 1.21 | 42.33 |
| | ACT | 44.18 | 0.40 | 1.25 | 43.60 |
| | CD | 433.94 | 1.49 | 3.17 | 57.82 |
| | GC | 29.30 | 0.37 | 1.68 | 43.71 |
| Random Forests | AF | 178.50 | 0.23 | 4.22 | 63.30 |
| | BH | 89.07 | 0.24 | 5.52 | 51.70 |
| | BS | 98.50 | 0.23 | 4.05 | 51.13 |
| | WQR | 298.46 | 0.29 | 9.35 | 63.12 |
| | ACT | 78.44 | 0.28 | 3.82 | 44.66 |
| | CD | 571.33 | 1.12 | 14.05 | 94.08 |
| | GC | 60.41 | 0.28 | 3.24 | 45.06 |
| Light-GBM | AF | 301.56 | 0.25 | 5.81 | 63.06 |
| | BH | 62.30 | 0.24 | 3.01 | 44.84 |
| | BS | 74.59 | 0.24 | 2.55 | 45.23 |
| | WQR | 361.19 | 0.29 | 5.68 | 58.97 |
| | ACT | 36.39 | 0.28 | 1.73 | 42.71 |
| | CD | 102.04 | 1.07 | 2.49 | 46.34 |
| | GC | 48.63 | 0.28 | 2.97 | 43.03 |
| Mean | | 148.92 | 0.41 | 3.94 | 51.44 |

Table 20: Computation time of different components of FEBP, in minutes.

| Model | Dataset | LLM Generation | Feature Evaluation | Feature Selection |
|---|---|---|---|---|
| Linear Model | AF | 16.73 | 22.98 | 3.08 |
| | BH | 18.50 | 20.18 | 2.60 |
| | WQR | 19.07 | 20.24 | 3.02 |
| | ACT | 18.92 | 20.97 | 3.71 |
| | CD | 16.73 | 25.14 | 15.95 |
| | GC | 17.01 | 23.24 | 3.47 |
| Random Forests | AF | 15.34 | 25.32 | 22.64 |
| | BH | 18.60 | 23.69 | 9.41 |
| | BS | 15.12 | 25.16 | 10.87 |
| | WQR | 12.75 | 23.81 | 26.56 |
| | ACT | 13.79 | 21.67 | 9.20 |
| | CD | 12.48 | 25.89 | 55.71 |
| | GC | 14.80 | 21.91 | 8.35 |
| Light-GBM | AF | 17.37 | 21.06 | 24.63 |
| | BH | 19.70 | 20.40 | 4.74 |
| | BS | 17.03 | 22.18 | 6.02 |
| | WQR | 16.27 | 21.19 | 21.51 |
| | ACT | 19.18 | 20.24 | 3.29 |
| | CD | 16.53 | 21.68 | 8.13 |
| | GC | 17.00 | 20.40 | 5.63 |
| Mean | | 16.65 | 22.37 | 12.43 |

## E ADDITIONAL ABLATION STUDY

To validate the effectiveness of our canonicalization scheme, we compare the full version of FEBP with the reduced version without RPN canonicalization. From Table 21, the full version outperforms the reduced version in terms of the mean performance score using all three downstream models. The Friedman test $p$-value is $1.29 \times 10^{-28}$. Table 22 summarizes the $p$-values from the Nemenyi post-hoc test for pairwise comparison, which shows that the performance difference is statistically significant at the $p = 0.05$ level for the cases with GPT-3.5 and post-AutoFE parameter tuning as well as GPT-4 without post-AutoFE parameter tuning. We also observe a decrease in the number of LLM responses without canonicalization. This is because the expression becomes more flexible, reducing the likelihood of duplication with existing features during feature generation.

Additionally, we find that when switching to prefix feature expressions, the LLM encounters difficulty generating syntactically valid feature expressions, leading to a failure to complete one single run in our experiments.

**Table 21:** Performance comparison of FEBP with and without RPN canonicalization. For each compared version, the left and middle columns show the results without and with parameter tuning of the downstream model algorithm post AutoFE, respectively, and the right column shows the number of LLM responses. The results where the full version outperforms the reduced version are highlighted in boldface.

| Model | Dataset | Raw | GPT-3.5 w/o Canonicalization | | | GPT-3.5 Full | | | GPT-4 w/o Canonicalization | | | GPT-4 Full | | |
|---|---|---|---|---|---|---|---|---|---|---|---|---|---|---|
| Linear Model | AF | 0.3474 | 0.6679 | 0.6688 | 338.6 | 0.6612 | 0.6616 | 339.8 | 0.6538 | 0.6529 | 321.2 | **0.6649** | **0.6647** | 371.4 |
| | BH | 0.3776 | 0.5048 | 0.5076 | 351.2 | 0.4995 | 0.5025 | 378.6 | 0.4987 | 0.5030 | 310.8 | **0.5184** | **0.5289** | 335.4 |
| | WQR | 0.2696 | 0.2702 | 0.2735 | 336.2 | **0.2722** | **0.2745** | 328.4 | 0.2690 | 0.2706 | 279.0 | **0.2713** | **0.2748** | 312.6 |
| | ACT | 0.8505 | 0.8748 | 0.8794 | 366.4 | 0.8729 | 0.8794 | 372.2 | 0.8738 | 0.8752 | 298.0 | **0.8766** | **0.8762** | 377.4 |
| | CD | 0.8267 | 0.8280 | 0.8290 | 350.4 | 0.8282 | 0.8282 | **342.0** | 0.8270 | 0.8271 | 285.4 | **0.8288** | **0.8288** | 250.4 |
| | GC | 0.7100 | 0.7370 | 0.7330 | 352.0 | **0.7570** | **0.7460** | 379.0 | 0.7550 | 0.7490 | 447.2 | **0.7590** | 0.7420 | 310.6 |
| Mean | | 0.5636 | 0.6471 | 0.6486 | 349.1 | **0.6485** | **0.6487** | 356.7 | 0.6462 | 0.6463 | 323.6 | **0.6532** | **0.6526** | 326.3 |
| Random Forests | AF | 0.7677 | 0.7628 | 0.7762 | 358.0 | **0.7709** | **0.7787** | 393.2 | 0.7743 | 0.7843 | 340.2 | 0.7681 | 0.7749 | **314.2** |
| | BH | 0.5426 | 0.5573 | 0.5573 | 364.0 | 0.5549 | 0.5533 | 374.4 | 0.5491 | 0.5460 | 322.4 | **0.5543** | **0.5522** | 278.6 |
| | BS | 0.9446 | 0.9804 | 0.9807 | 372.2 | **0.9873** | **0.9881** | 386.8 | 0.9778 | 0.9777 | 284.4 | **0.9845** | **0.9848** | 255.0 |
| | WQR | 0.3662 | 0.3776 | 0.3726 | 334.6 | **0.3862** | **0.3845** | 362.6 | 0.3739 | 0.3719 | 269.8 | **0.3810** | **0.3810** | 283.2 |
| | ACT | 0.8808 | 0.8879 | 0.8841 | 353.4 | **0.8925** | 0.8921 | 357.6 | 0.8841 | 0.8864 | 327.6 | **0.8893** | 0.8864 | 424.0 |
| | CD | 0.8293 | 0.8283 | 0.8285 | 381.6 | **0.8295** | 0.8294 | 349.8 | 0.8290 | 0.8287 | 297.2 | **0.8295** | 0.8276 | 304.0 |
| | GC | 0.7450 | 0.7660 | 0.7620 | 342.2 | 0.7640 | 0.7620 | 368.2 | 0.7680 | 0.7610 | 368.2 | **0.7680** | **0.7680** | 471.8 |
| Mean | | 0.6806 | 0.7372 | 0.7373 | 358.0 | **0.7408** | **0.7412** | 370.4 | 0.7366 | 0.7366 | 315.7 | **0.7392** | **0.7393** | 333.0 |
| Light-GBM | AF | 0.8375 | 0.8322 | 0.8365 | 343.6 | 0.8311 | **0.8392** | 380.2 | 0.8280 | 0.8350 | 376.0 | **0.8366** | **0.8395** | 360.6 |
| | BH | 0.5537 | 0.5599 | 0.5556 | 339.2 | **0.5619** | **0.5644** | 342.0 | 0.5577 | 0.5548 | 315.2 | **0.5642** | **0.5595** | 345.6 |
| | BS | 0.9429 | 0.9643 | 0.9664 | 368.8 | **0.9737** | **0.9754** | 380.0 | 0.9597 | 0.9609 | 276.2 | **0.9801** | **0.9813** | 236.8 |
| | WQR | 0.3825 | 0.4075 | 0.4042 | 346.4 | **0.4118** | **0.4171** | 322.8 | 0.4036 | 0.4032 | 288.2 | 0.4021 | **0.4042** | 293.6 |
| | ACT | 0.8832 | 0.8813 | 0.8860 | 342.4 | **0.8888** | **0.8925** | 367.4 | 0.8822 | 0.8879 | 313.2 | **0.8902** | **0.8925** | 359.6 |
| | CD | 0.8300 | 0.8302 | 0.8291 | 355.8 | 0.8301 | 0.8297 | 352.2 | 0.8295 | 0.8291 | 301.6 | **0.8303** | **0.8294** | 371.2 |
| | GC | 0.7250 | 0.7640 | 0.7650 | 346.2 | **0.7680** | **0.7720** | 376.6 | 0.7620 | 0.7650 | 428.8 | **0.7760** | **0.7700** | 382.2 |
| Mean | | 0.6806 | 0.7485 | 0.7490 | 348.9 | **0.7522** | **0.7558** | 360.2 | 0.7461 | 0.7480 | 328.5 | **0.7542** | **0.7538** | 335.7 |
| Mean | | 0.6806 | 0.7141 | 0.7148 | 352.2 | **0.7171** | **0.7185** | 362.7 | 0.7128 | 0.7135 | 322.5 | **0.7187** | 0.7183 | 331.9 |

**Table 22:** The Nemenyi post-hoc test $p$-values for pairwise comparison of the methods in Table 21. Results that are significant at the $p = 0.05$ confidence level are highlighted in boldface.

| | | Raw | GPT-3.5 w/o | | Full | | GPT-4 w/o | | Full | |
|---|---|---|---|---|---|---|---|---|---|---|
| Raw | | 1.0000 | **0.0010** | **0.0010** | **0.0010** | **0.0010** | **0.0010** | **0.0010** | **0.0010** | **0.0010** |
| GPT-3.5 | w/o | **0.0010** | 1.0000 | 0.9000 | 0.2977 | **0.0060** | 0.9000 | 0.9000 | **0.0224** | 0.4293 |
| | | **0.0010** | 0.9000 | 1.0000 | 0.6618 | **0.0433** | 0.8811 | 0.8889 | 0.1230 | 0.7871 |
| | Full | **0.0010** | 0.2977 | 0.6618 | 1.0000 | 0.9000 | **0.0341** | **0.0355** | 0.9000 | 0.9000 |
| | | **0.0010** | **0.0060** | **0.0433** | 0.9000 | 1.0000 | **0.0010** | **0.0010** | 0.9000 | 0.8028 |
| GPT-4 | w/o | **0.0010** | 0.9000 | 0.8811 | **0.0341** | **0.0010** | 1.0000 | 0.9000 | **0.0010** | 0.0635 |
| | | **0.0010** | 0.9000 | 0.8889 | **0.0355** | **0.0010** | 0.9000 | 1.0000 | **0.0010** | 0.0659 |
| | Full | **0.0010** | **0.0224** | 0.1230 | 0.9000 | 0.9000 | **0.0010** | **0.0010** | 1.0000 | 0.9000 |
| | | **0.0010** | 0.4293 | 0.7871 | 0.9000 | 0.8028 | 0.0635 | 0.0659 | 0.9000 | 1.0000 |

# F  ADDITIONAL ANALYSIS

## F.1  FEATURE ANALYSIS

Figure 13 compares the proportions of generated features selecting each feature attribute across different datasets and downstream models (linear models and Random Forests) for both the full and semantically blinded versions of FEBP. In the blinded version, we observe that the LLM tends to prioritize earlier feature attributes in the dataset while paying less attention to later ones, reflecting an inherent bias of the language model. In contrast, in the full version, the selection of feature attributes is guided by the semantic information of the dataset rather than the positional order of the attributes. Specifically, Attribute 19 *CD4 at baseline* in AIDS Clinical Trials (ACT) and Attribute 10 *alcohol* in Wine Quality Red (WQR), which contain useful information for predicting the targets *censoring indicator* and *quality*, respectively, are included in the majority of the generated features. This demonstrates the role of dataset semantic information in the LLM-based feature search process.

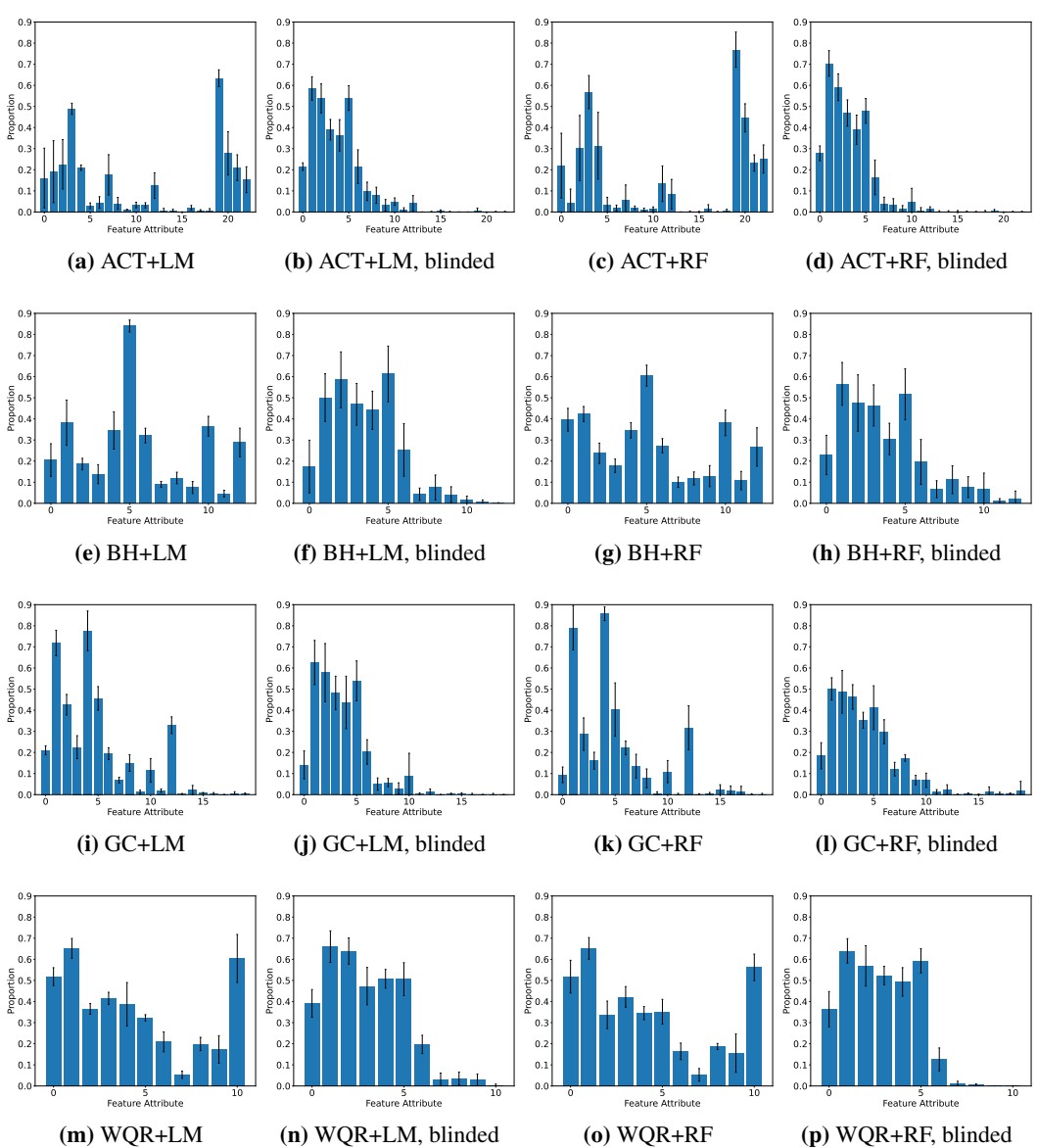

**(a)** ACT+LM    **(b)** ACT+LM, blinded    **(c)** ACT+RF    **(d)** ACT+RF, blinded

**(e)** BH+LM    **(f)** BH+LM, blinded    **(g)** BH+RF    **(h)** BH+RF, blinded

**(i)** GC+LM    **(j)** GC+LM, blinded    **(k)** GC+RF    **(l)** GC+RF, blinded

**(m)** WQR+LM    **(n)** WQR+LM, blinded    **(o)** WQR+RF    **(p)** WQR+RF, blinded

**Figure 13:** The proportions of generated features selecting each feature attribute in the dataset.

## F.2 FEATURE IMPORTANCE

Figure 14 shows the feature importance across different datasets and downstream models. For linear models, we use the magnitudes of coefficients; for Random Forests (Breiman, 2001), we use the impurity-based feature importance; for LightGBM (Ke et al., 2017), we use the total gains of splits. FEBP enhances the datasets by constructing new features that provide valuable information for predicting the target. Additionally, we observe that Random Forests and LightGBM benefit from more complex features compared to linear models, as they are capable of synthesizing features internally. Our method adaptively adjusts the feature complexity to suit different downstream models.

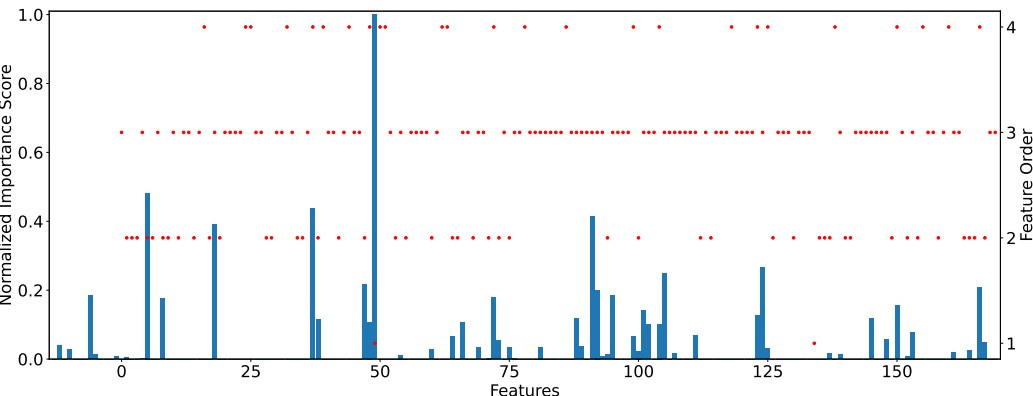

**(a)** BH+LM. Test performance improves from to 0.3776 to 0.5157.

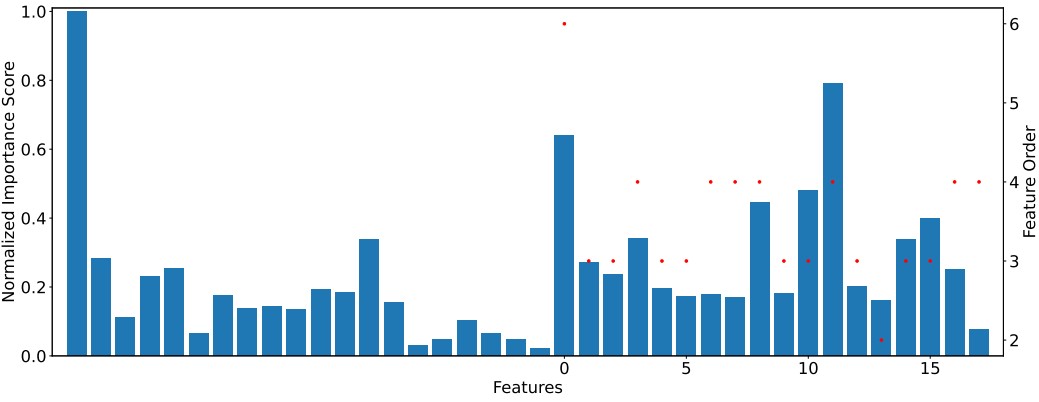

**(b)** GC+RF. Test performance improves from to 0.7450 to 0.7700.

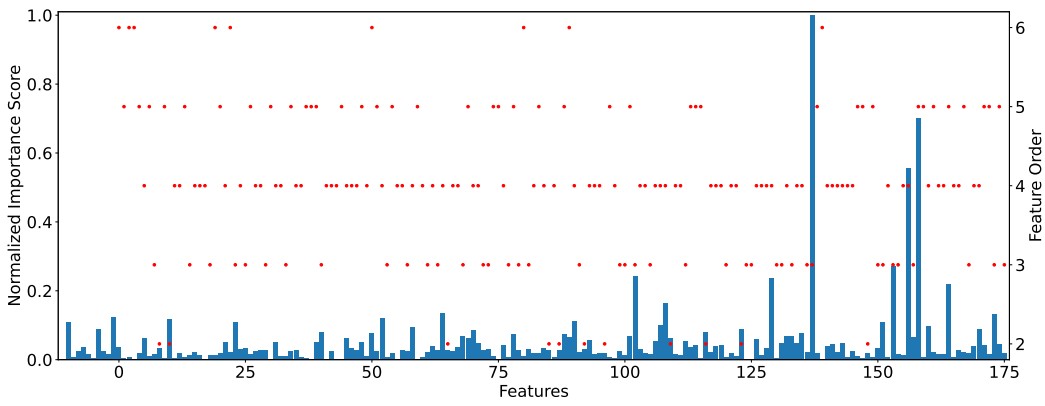

**(c)** WQR+LGBM. Test performance improves from to 0.3825 to 0.4299.

**Figure 14:** The blue bars show the normalized feature importance scores. The red dots show the order of features. The generated features are positioned on the x-axis starting at index 0, following the raw features.

### F.3 FEATURE SEARCH EXAMPLE

Figure 15 shows samples of the LLM-based feature search process from a single run. In iteration 155, the generated feature is derived from the combination of three example features. In iteration 158, the generated feature is derived from the combination of two example features, with the feature attributes modified. Iteration 160 exhibits an instance of modification on the example feature. Iteration 161 exhibits an instance of analogy of the example feature.

Figure 15: Examples of feature search. The prompt examples that relate to the generated feature are highlighted in red.

Iteration 155
Generated feature:
6, 12, division, 18, *, min_max
Prompt examples:
4, log, 13, *, 17, min_max, sqrt_abs, +
6, 12, *, 21, sqrt_abs, +
4, 14, +, 18, *, min_max
11, 21, +, 6, *, sqrt_abs
2, 4, +, 6, division, log
4, 6, +, 16, *, 17, +, min_max
6, 21, division, 17, log, *
12, min_max, 18, *, 5, mod_column
2, 6, mod_column, 17, +, sqrt_abs
2, 3, +, 4, sqrt_abs, *
4, 12, division, 16, *, min_max, sqrt_abs

Iteration 158
Generated feature:
5, log, 19, *, 15, +, min_max
Prompt examples:
4, log, 13, *, 17, min_max, sqrt_abs, +
6, 12, *, 21, sqrt_abs, +
4, 14, +, 18, *, min_max
11, 21, +, 6, *, sqrt_abs
2, 4, +, 6, division, log
4, 6, +, 16, *, 17, +, min_max
6, 21, division, 17, log, *
12, min_max, 18, *, 5, mod_column
2, 6, mod_column, 17, +, sqrt_abs
2, 3, +, 4, sqrt_abs, *
4, 12, division, 16, *, min_max, sqrt_abs

Iteration 160
Generated feature:
11, 21, *, 4, +, 14, mod_column
Prompt examples:
4, log, 13, *, 17, min_max, sqrt_abs, +
6, 12, *, 21, sqrt_abs, +
4, 14, +, 18, *, min_max
11, 21, +, 6, *, sqrt_abs
2, 4, +, 6, division, log
4, 6, +, 16, *, 17, +, min_max
6, 21, division, 17, log, *
12, min_max, 18, *, 5, mod_column
2, 6, mod_column, 17, +, sqrt_abs
2, 3, +, 4, sqrt_abs, *
4, 12, division, 16, *, min_max, sqrt_abs

Iteration 161
Generated feature:
6, 12, +, 17, *, min_max
Prompt examples:

```
6, 12, *, 21, sqrt_abs, +
4, 14, +, 18, *, min_max
11, 21, *, 4, +, 14, mod_column
11, 21, +, 6, *, sqrt_abs
2, 4, +, 6, division, log
4, 6, +, 16, *, 17, +, min_max
6, 21, division, 17, log, *
12, min_max, 18, *, 5, mod_column
2, 6, mod_column, 17, +, sqrt_abs
2, 3, +, 4, sqrt_abs, *
4, 12, division, 16, *, min_max, sqrt_abs
```

# G   DIFFERENCES TO CAAFE

While our work FEBP and CAAFE (Hollmann et al., 2023) both utilize LLMs to construct new features incorporating dataset semantic information, they differ in several key aspects. We design FEBP such that it taps into the in-context learning capability of LLMs and performs effective feature search. In FEBP, we provide top-performing constructed features in the prompt as learning examples, label them with performance scores, and rank them by score. We demonstrate that the LLM learns to optimize feature construction over the course of algorithm. CAAFE instead stores all previous instructions and code snippets in the conversation history, which hinders the in-context learning of optimal feature patterns. It quickly consumes the LLM's context as the algorithm iterates, incurring more and more LLM generation costs. In comparison, the LLM generation cost of FEBP stays constant across iterations, without a maximum limit on the number of iterations it can perform. Therefore, our method FEBP has stronger capability of performing feature search in large search spaces requiring many iterations, such as datasets with numerous feature attributes.

In FEBP, we also explore representing features in a different form, i.e., canonical RPN (cRPN). We refer to Appendix A for further detail. Compared with the Python code representation in CAAFE, cRPN is more compact, which not only reduces LLM generation costs but also makes the in-context learning of feature patterns easier, and more human interpretable. The use of pre-defined operators reduces the search space and simplifies the learning process for optimizing feature construction. Together, our approach gives better control than code representation and helps avoid undesirable or unexpected LLM outputs. Another advantage of cRPN is that it is convenient to import external features (as outlined in Algorithm 1) and export the results as individual features, providing compatibility with other feature engineering methods.

More fundamentally, we demonstrate in this work that general-purpose LLMs like GPTs can effectively model recursive tree structures in the form of cRPN feature expressions and reason about the structures in the context of semantic information, paving the way for further LLM-driven applications. We hereby underscore the importance of adopting proper representation for the downstream task to tap into LLMs' potential.

