# OpenReview forum: "Automated Feature Engineering by Prompting"
_ICLR.cc/2025/Conference — Submitted to ICLR 2025_

### Official Review · Reviewer_nSbg · 2024-11-03

**Soundness:** 3
**Presentation:** 3
**Contribution:** 2
**Rating:** 5
**Confidence:** 4

**Summary:**

Automated feature engineering (AutoFE) uses high-level algorithms and models to automate the FE
process such that the performance is comparable to domain experts. Existing AutoFE methods compute
and evaluate a large number of features in a trial-and-error manner. The authors propose a method that
leverages LLMs to do 'effective, efficient, and interpretable feature engineering.' Their main contributions
are to generate features in string representations while providing semantic explanations, to benchmark
the performance of the approach against state-of-the-art baselines using both GPT-3.5 and GPT-
4, and to investigate the impact of semantic context on this approach.

**Strengths:**

--The problem is obviously an interesting one, and the authors write it up with sufficient clarity. The paper is very readable.
--There is some attempt at formalism, which I do appreciate. Far too many papers only describe a problem 'intuitively', and this is usually not adequate for a scientific venue in my opinion.
--In some cases, the experimental improvements are decent, and I am particularly appreciative that the authors put in significance results.

**Weaknesses:**

There are two main weaknesses. The first is simply the nature of the contribution: I am not convinced that there is anything significantly novel or surprising here. As the authors state, LLMs have a lot of background knowledge that can be used to generate useful features, and the second 'self-correction' mechanism that subsets constructed features based on performance evaluation is not new either. It has been used a lot in other such LLM-related works, such as in the text2sql problem. I took a close look at the algorithm that the authors put on page 5 but it turns out that it's just a long way of expressions the rather simple formulation in figure 1.

A much more significant weakness is in the experimental section. In carefully looking at the results, I am seeing hardly any difference between the authors' approach and baselines when using classifiers like Light-GBM and random forests. The only real difference seems to be in linear models, and I don't see why on the basis of this one model we should give the authors' approach too much credit.

**Questions:**

None.

---

> ### Author Response · Authors · 2024-11-25
> **Rebuttal by Authors**
>
> Dear Reviewer nSbg,
>
> Thank you for acknowledging the importance of our research problem and the clarity of the paper.
> We are pleased that you like our attempt at formalism.
> We next address the highlighted weaknesses to improve our paper's quality and clarity.
>
> ---
> > The nature of the contribution.
>
> We propose a novel approach to LLM-based automated feature engineering
> that addresses the weaknesses of the state-of-the-art approach CAAFE [Hollmann2023].
> We add a discussion on our contributions in **new Section G**.
> We highlight the key differences between our work FEBP and CAAFE as follows
> 1. In FEBP, we provide top-performing constructed features in the prompt as learning examples, label them with performance scores, and rank them by score.
> CAAFE instead stores all previous instructions and code snippets in the conversation history.
> The advantages of our approach include
>     - More effective **in-context learning of feature patterns**.
>     - The LLM generation cost **stays constant** across feature search iterations, without consuming extra LLM context.
>     - Stronger capability of performing feature search in **large search spaces** by enabling more iterations, providing pronounced benefits in presence of numerous feature attributes.
> 2. In FEBP, we represent features in the form of canonical RPN (cRPN), while CAAFE represents features with Python code.
> The advantages of our approach include
>     - cRPN is more compact, **reducing LLM generation costs**.
>     - The compactness of cRPN makes **in-context learning of feature patterns** easier.
>     - The use of pre-defined operators **reduces the search space** and **simplifies the learning process** for optimizing feature construction.
>     - Our approach gives **better control**, such as the number of operators to use in features, and helps avoid undesirable or unexpected LLM outputs, such as malicious code snippets.
>     - It is convenient to import external features and export the results as individual features, providing **compatibility with other AutoFE methods**.
> 3. We demonstrate in this work that general-purpose LLMs like GPTs can effectively model **recursive tree structures** in the form of cRPN feature expressions and reason about the structures in the context of **semantic information**.
>
> We hope this explanation makes the contributions of our work clearer.
>
> [Hollmann2023] Large language models for automated data
> science: Introducing CAAFE for context-aware automated feature engineering. NIPS, 2023

---

> ### Author Response · Authors · 2024-11-25
> **Rebuttal by Authors - Continued**
>
> > Weakness in the experimental section.
>
> We have examined the impact of dataset semantic context on performance and feature construction efficiency in **Section 5.3**.
> We have provided detailed analyses on the LLM-based feature search process in **Section 5.4** and the effects of hyperparameters in **Section 5.5**.
>
> We add an additional ablation study regarding RPN canonicalization in **new Section E**.
> We summarize the new experimental results as follows
>
> |         | GPT-3.5 |        |       |            |        |       | GPT-4  |        |       |        |        |       |
> |---------|---------|--------|-------|------------|--------|-------|--------|--------|-------|--------|--------|-------|
> |         | w/o     |        |       | Full       |        |       | w/o    |        |       | Full   |        |       |
> | Linear  | 0.6471  | 0.6486 | 349.1 | **0.6485** | **0.6487** | 356.7 | 0.6462 | 0.6463 | 323.6 | **0.6532** | **0.6526** | 326.3 |
> | RF      | 0.7372  | 0.7373 | 358.0 | **0.7408** | **0.7412** | 370.4 | 0.7366 | 0.7366 | 315.7 | **0.7392** | **0.7393** | 333.0 |
> | LGBM    | 0.7485  | 0.7490 | 348.9 | **0.7522** | **0.7558** | 360.2 | 0.7461 | 0.7480 | 328.5 | **0.7542** | **0.7538** | 335.7 |
> | Mean    | 0.7141  | 0.7148 | 352.2 | **0.7171** | **0.7185** | 362.7 | 0.7128 | 0.7135 | 322.5 | **0.7187** | **0.7183** | 331.9 |
>
> **Without RPN canonicalization**, we observe performance degradation using all three downstream models.
> The Friedman-Nemenyi post-hoc test shows that the performance difference is statistically significant at the p=0.05 level
> for the cases with GPT-3.5 and post-AutoFE parameter tuning as well as GPT-4 without post-AutoFE parameter tuning.
> We also observe a decrease in the number of LLM responses without canonicalization. This is because the expression becomes more flexible, reducing the likelihood of duplication with existing features.
> Additionally, we find that when switching to **prefix feature expressions**, the LLM encounters difficulty generating syntactically valid feature expressions, leading to a failure to complete one single run in our experiments.
>
> Additionally, we add a discussion on why we use canonical RPN in **new Section A**.
> We represent features in **RPN** because it is compact and unambiguous and it better encodes the recursive structure for sequential modeling.
> We **canonicalize RPN** because that ensures the consistency of feature representations and facilitates the in-context learning of feature patterns.
> Our canonicalization scheme also introduces left skewness to the expression tree that enhances the clarity of the recursive structure in cRPN.
>
> Furthermore, we add an additional feature analysis in **new Section F.1**.
> We compare the proportions of constructed features selecting each feature attribute in the datasets for both the full and semantically blinded versions of FEBP.
> In the **blinded version**, we observe that the LLM tends to prioritize earlier feature attributes in the dataset while paying less attention to later ones, reflecting an inherent bias of the language model.
> In contrast, in the **full version**, the selection of feature attributes is guided by the semantic information of the dataset rather than the positional order of the attributes.
> This demonstrates the role of **dataset semantic information** in the LLM-based feature construction process.
>
> We also add an additional analysis of feature importance in **new Section F.2**.
> We find that the features constructed by our method have high importance in the downstream models.
> Moreover, our method adaptively adjusts the feature complexity to suit different downstream models.

---

> ### Author Response · Authors · 2024-11-26
> **Rebuttal by Authors - Continued2**
>
> > In carefully looking at the results, I am seeing hardly any difference between the authors' approach and baselines when using classifiers like Light-GBM and random forests.
>
> We compute the mean percentage performance improvement of our method FEBP over the baseline methods with GPT-3.5 and GPT-4, respectively.
> We add the full results to our revised version as **Tables 8 and 9 in New Section D.5**.
>
> With GPT-3.5,
>
> |        | Raw   |       | DIFER |       | OpenFE |       | CAAFE |       |
> |--------|-------|-------|-------|-------|--------|-------|-------|-------|
> | Linear | 22.17 | 22.33 | 4.76  | 2.36  | 14.30  | 14.50 | 12.35 | 12.50 |
> | **RF**    | 2.36  | 2.37  | 0.02  | -0.08 | 0.49   | 0.63  | 0.54  | 0.67  |
> | **LGBM**    | 2.60  | 3.16  | 0.18  | 0.32  | 1.21   | 1.58  | 0.82  | 1.45  |
> | Mean   | 8.39  | 8.63  | 1.50  | 0.79  | 4.88   | 5.12  | 4.18  | 4.49  |
>
> With GPT-4,
>
> |        | Raw   |       | DIFER |       | OpenFE |       | CAAFE |       |
> |--------|-------|-------|-------|-------|--------|-------|-------|-------|
> | Linear | 23.26 | 23.52 | 5.56  | 3.21  | 15.33  | 15.64 | 12.17 | 12.33 |
> | **RF**      | 2.08  | 2.07  | -0.26 | -0.37 | 0.21   | 0.33  | 0.69  | 0.76  |
> | **LGBM**    | 2.67  | 2.60  | 0.27  | -0.19 | 1.27   | 1.03  | 2.01  | 1.15  |
> | Mean   | 8.64  | 8.69  | 1.67  | 0.76  | 5.12   | 5.17  | 4.60  | 4.37  |
>
>
> To examine the performance difference when using Random Forests and LightGBM, we perform additional statistical tests excluding linear model results and add the new results to **Section D.7**.
> The p-values from the Nemenyi post-hoc test are summarized as follows
>
> |          |         | Raw        | DIFER      |        | OpenFE |        | CAAFE   |        |        |        |
> |----------|---------|------------|------------|--------|--------|--------|---------|--------|--------|--------|
> |          |         |            |            |        |        |        | GPT-3.5 |        | GPT-4  |        |
> | **FEBP** | GPT-3.5 | **0.0010** | 0.4263     | 0.9000 | **0.0010** | **0.0010** | 0.4493  | 0.3422 | **0.0010** | **0.0026** |
> |          |         | **0.0010** | **0.0299** | 0.8377 | **0.0010** | **0.0010** | **0.0334**  | **0.0199** | **0.0010** | **0.0010** |
> |          | GPT-4   | **0.0010** | 0.1392     | 0.9000 | **0.0010** | **0.0010** | 0.1516  | 0.1017 | **0.0010** | **0.0010** |
> |          |         | **0.0010** | 0.9000     | 0.9000 | **0.0171** | **0.0289** | 0.9000  | 0.9000 | **0.0299** | 0.0661 |
>
>
> We note that FEBP with GPT-3.5 and post-AutoFE parameter tuning significantly outperforms all baselines except DIFER at the p = 0.05 level.
> With GPT-4, the performance difference between FEBP and CAAFE is statistically significant at the p = 0.05 level.
> We have highlighted the corresponding parts of **Table 16** in our revised version for better clarity.
>
> We hope this better shows the performance improvement of our approach
> over the baselines when using Light-GBM and random forests.
>
> We would like to note that the reason why the performance difference is less salient when using Light-GBM and random forests
> is because these models can capture complex non-linear relationships from raw features themselves, which dilutes the effect of feature engineering.
> Thus, in our view, the performance improvement when using simple downstream models like linear models better demonstrates the efficacy of AutoFE.

---

> > ### Author Response · Authors · 2024-12-01
> > **Followup on Rebuttal Phase Feedback**
> >
> > Dear Reviewer nSbg,
> >
> > Thank you for reviewing our paper and providing such valuable and constructive feedback. We have carefully studied your suggestions and made several revisions, adding extensive experimental details to enhance the paper’s clarity, depth, and contributions. Your insightful comments have been instrumental in guiding these improvements.
> >
> > As the rebuttal phase is nearing its deadline, we are looking forward to engaging in a timely discussion.
> > If you have any further questions or concerns, please do not hesitate to let us know. We are more than willing to provide additional clarifications and supporting materials to improve the quality of our work.
> >
> > Additionally, we kindly hope that these updates and clarifications will encourage you to reconsider your evaluation, as they directly address your constructive feedback.
> >
> > Thank you again for your invaluable time and effort!
> >
> > Best regards,
> >
> > Authors

---

### Official Review · Reviewer_gWBg · 2024-11-03

**Soundness:** 3
**Presentation:** 3
**Contribution:** 2
**Rating:** 5
**Confidence:** 3

**Summary:**

This paper propose a novel AutoFE algorithm that leverage LLMs to automatically process dataset descriptions and generate new features. Specifically, LLM produce some feature with the semantic information of dataset and then iteratively refines feature construction with new output example features. This method of using LLMs not only reduces manual effort, but also proves in experiments that its effect reaches SOTA in AutoFE.

**Strengths:**

1. Experiments are tested on multiple real-world datasets (or tasks)

2. The writing of the paper is complete and clear, making it easy to read.

**Weaknesses:**

1. The new feature space generated by this method is limited, and it can only generate new features based on given operators.

2. The experimental conclusions in Table 2 are not consistent. The results of GPT3.5 are better than those of GPT4, which is counterintuitive. I think an additional experiment is needed to reveal the reason for this, and the speculation in line 320 does not explain the reason for this phenomenon.

3. Innovation is limited. In my opinion, this method is a prompting method designed based on LLMs. The entire framework implements automated FE through LLM simulation experts but lacks more detailed analysis and module design. For example, I think the analysis of temperature is redundant.

**Questions:**

See above.

---

> ### Author Response · Authors · 2024-11-25
> **Rebuttal by Authors**
>
> Dear Reviewer gWBg,
>
> Thank you for acknowledging our selection of datasets and tasks and the clarity of the paper.
> We want to address the highlighted weaknesses and answer the posed questions to improve our paper's quality and clarity.
> Next we reply to your comments.
>
> ---
> > The new feature space generated by this method is limited, and it can only generate new features based on given operators.
>
> In this work, we explore an alternative to representation learning for automatically extracting feature information from datasets, i.e., AutoFE, which is a fundamental aspect of data-centric AI.
> AutoFE is particularly well-suited for **small datasets**, where the limited data may be insufficient to train a deep neural network for representation learning.
> Our LLM-based AutoFE approach generates meaningful features by performing mathematical feature-feature crossing while also providing semantic explanations.
> The use of pre-defined transformation operators enhances both the **controllability** of the feature generation process and the **interpretability** of the extracted features.
>
> We select the same set of feature transformation operators as those in DIFER [Zhu22] to ensure a fair comparison.
> It is worth noting that the set of operators in our method is easily **extendable**, allowing for improved performance by generating features within a broader search space.
>
> It is impossible to conduct feature engineering without a give set of operators.
> Even if features are specified by code, it still means a finite set of operators.
>
> [Zhu22] DIFER: Differentiable automated feature engineering. The first International Conference on Automated Machine
> Learning, 2022
>
> ---
>
> > Innovation is limited.
>
> We propose a novel approach to LLM-based automated feature engineering
> that addresses the weaknesses of the state-of-the-art approach CAAFE [Hollmann2023].
> We add a discussion on the novelty in **new Section G**.
> We highlight the key differences between our work FEBP and CAAFE as follows
> 1. In FEBP, we provide top-performing constructed features in the prompt as learning examples, label them with performance scores, and rank them by score.
> CAAFE instead stores all previous instructions and code snippets in the conversation history.
> The advantages of our approach include
>     - More effective **in-context learning of feature patterns**.
>     - The LLM generation cost **stays constant** across feature search iterations, without consuming extra LLM context.
>     - Stronger capability of performing feature search in **large search spaces** by enabling more iterations, providing pronounced benefits in presence of numerous feature attributes.
> 2. In FEBP, we represent features in the form of canonical RPN (cRPN), while CAAFE represents features with Python code.
> The advantages of our approach include
>     - cRPN is more compact, **reducing LLM generation costs**.
>     - The compactness of cRPN makes **in-context learning of feature patterns** easier.
>     - The use of pre-defined operators **reduces the search space** and **simplifies the learning process** for optimizing feature construction.
>     - Our approach gives **better control**, such as the number of operators to use in features, and helps avoid undesirable or unexpected LLM outputs, such as malicious code snippets.
>     - It is convenient to import external features and export the results as individual features, providing **compatibility with other AutoFE methods**.
> 3. We demonstrate in this work that general-purpose LLMs like GPTs can effectively model **recursive tree structures** in the form of cRPN feature expressions and reason about the structures in the context of **semantic information**.
>
> We hope this explanation makes the contributions of our work clearer.
>
> [Hollmann2023] Large language models for automated data
> science: Introducing CAAFE for context-aware automated feature engineering. NIPS, 2023

---

> ### Author Response · Authors · 2024-11-25
> **Rebuttal by Authors - Continued**
>
> > Lacks more detailed analysis and module design.
>
> We have examined the impact of dataset semantic context on performance and feature construction efficiency in **Section 5.3**.
> We have provided detailed analyses on the LLM-based feature search process in **Section 5.4** and the effects of hyperparameters in **Section 5.5**.
>
> We add a discussion on why we use canonical RPN in **new Section A**.
> We represent features in **RPN** because it is compact and unambiguous and it better encodes the recursive structure for sequential modeling.
> We **canonicalize RPN** because that ensures the consistency of feature representations and facilitates the in-context learning of feature patterns.
> Our canonicalization scheme also introduces left skewness to the expression tree that enhances the clarity of the recursive structure in cRPN.
>
> We add an additional ablation study regarding RPN canonicalization in **new Section E**.
> We summarize the new experimental results as follows
>
> |         | GPT-3.5 |        |       |            |        |       | GPT-4  |        |       |        |        |       |
> |---------|---------|--------|-------|------------|--------|-------|--------|--------|-------|--------|--------|-------|
> |         | w/o     |        |       | Full       |        |       | w/o    |        |       | Full   |        |       |
> | Linear  | 0.6471  | 0.6486 | 349.1 | **0.6485** | **0.6487** | 356.7 | 0.6462 | 0.6463 | 323.6 | **0.6532** | **0.6526** | 326.3 |
> | RF      | 0.7372  | 0.7373 | 358.0 | **0.7408** | **0.7412** | 370.4 | 0.7366 | 0.7366 | 315.7 | **0.7392** | **0.7393** | 333.0 |
> | LGBM    | 0.7485  | 0.7490 | 348.9 | **0.7522** | **0.7558** | 360.2 | 0.7461 | 0.7480 | 328.5 | **0.7542** | **0.7538** | 335.7 |
> | Mean    | 0.7141  | 0.7148 | 352.2 | **0.7171** | **0.7185** | 362.7 | 0.7128 | 0.7135 | 322.5 | **0.7187** | **0.7183** | 331.9 |
>
> **Without RPN canonicalization**, we observe performance degradation using all three downstream models.
> The Friedman-Nemenyi post-hoc test shows that the performance difference is statistically significant at the p=0.05 level
> for the cases with GPT-3.5 and post-AutoFE parameter tuning as well as GPT-4 without post-AutoFE parameter tuning.
> We also observe a decrease in the number of LLM responses without canonicalization. This is because the expression becomes more flexible, reducing the likelihood of duplication with existing features.
> Additionally, we find that when switching to **prefix feature expressions**, the LLM encounters difficulty generating syntactically valid feature expressions, leading to a failure to complete one single run in our experiments.
>
> Furthermore, we add an additional feature analysis in **new Section F.1**.
> We compare the proportions of constructed features selecting each feature attribute in the datasets for both the full and semantically blinded versions of FEBP.
> In the **blinded version**, we observe that the LLM tends to prioritize earlier feature attributes in the dataset while paying less attention to later ones, reflecting an inherent bias of the language model.
> In contrast, in the **full version**, the selection of feature attributes is guided by the semantic information of the dataset rather than the positional order of the attributes.
> This demonstrates the role of **dataset semantic information** in the LLM-based feature construction process.
>
> We also add an additional analysis of feature importance in **new Section F.2**.
> We find that the features constructed by our method have high importance in the downstream models.
> Moreover, our method adaptively adjusts the feature complexity to suit different downstream models.
>
> ---
> > I think the analysis of temperature is redundant.
>
> The sampling temperature of the LLM is a relevant hyperparameter in our method, the other relevant one being the number of example features in the prompt.
> As discussed in **lines 252-254** of our paper:
> ```
> The sampling temperature of the LLM can be adjusted to balance exploration and exploitation, with higher temperatures encouraging more diverse solutions and lower temperatures favoring incremental changes to existing examples.
> ```
> In essence, the temperature functions similarly to the learning rate in optimization algorithms.
> From our experimental results presented in **Table 4**, the temperature impacts both performance and feature construction efficiency of our method with [statistical significance](https://openreview.net/forum?id=ZXO7iURZfW&noteId=E9VJAUTBgO).
> We select the optimal temperature (1) based on validation performance for achieving the best test results.

---

> > ### Author Response · Authors · 2024-12-01
> > **Followup on Rebuttal Phase Feedback**
> >
> > Dear Reviewer gWBg,
> >
> > Thank you for reviewing our paper and providing such valuable and constructive feedback. We have carefully studied your suggestions and made several revisions, adding extensive experimental details to enhance the paper’s clarity, depth, and contributions. Your insightful comments have been instrumental in guiding these improvements.
> >
> > As the rebuttal phase is nearing its deadline, we are looking forward to engaging in a timely discussion.
> > If you have any further questions or concerns, please do not hesitate to let us know. We are more than willing to provide additional clarifications and supporting materials to improve the quality of our work.
> >
> > Additionally, we kindly hope that these updates and clarifications will encourage you to reconsider your evaluation, as they directly address your constructive feedback.
> >
> > Thank you again for your invaluable time and effort!
> >
> > Best regards,
> >
> > Authors

---

> ### Author Response · Authors · 2024-12-04
> **Rebuttal by Authors - Continued2**
>
> > The speculation in line 320 does not explain the reason for this phenomenon.
>
> To verify our speculation in line 320, we conduct an additional experiment with GPT-4 by varying the number of example
> features in the prompt. We summarize the validation performance as follows
>
> | Model    | Dataset | 1      | 5      | 10     | 20     | 30     |
> |----------|---------|--------|--------|--------|--------|--------|
> | RF       | AF      | 0.7864 | 0.7922 | 0.7905 | 0.7897 | 0.7920 |
> |          | WQR     | 0.3847 | 0.3835 | 0.3839 | 0.3862 | 0.3862 |
> |          | CD      | 0.8219 | 0.8218 | 0.8218 | 0.8219 | 0.8222 |
> | LGBM     | AF      | 0.8387 | 0.8413 | 0.8401 | 0.8433 | 0.8411 |
> |          | WQR     | 0.4216 | 0.4242 | 0.4290 | 0.4258 | 0.4267 |
> |          | CD      | 0.8231 | 0.8234 | 0.8227 | 0.8229 | 0.8231 |
> | **Mean** |         | 0.6794 | 0.6810 | 0.6813 | 0.6816 | 0.6819 |
>
> We observe improved performance as the number of example features increases.
> This indicates that incorporating **more example features** helps fully leverage GPT-4's enhanced in-context
> learning capabilities. We will include these results in the paper.
> For our evaluations, we set the number of example features to 10 for a fair comparison with GPT-3.5.

---

### Official Review · Reviewer_ogYM · 2024-11-04

**Soundness:** 3
**Presentation:** 3
**Contribution:** 3
**Rating:** 6
**Confidence:** 3

**Summary:**

The paper investigates the application of prompting LLM for automatic feature engineering over structured (tabular) data. It proposes an iterative in context learning approach (FEBP), which derives new features from raw tabular data. The iteration first leverages metadata and examples to generate new features, which are subsequently evaluated and added to the prompt, if they perform well. The approach stops if the model performance does not improve. FEBP is based on GPT-[3.5,4]. It is evaluated on 3 classification and 4 regression datasets against three automatic feature engineering methods.

**Strengths:**

The paper presents a sound approach for encoding tabular data for prompting and deriving new features. The approach is easy to replicate and can be reused quickly to train downstream models. Compared to deep embedding based approaches, the feature and their interactions are still explainable to a certain extent (with some effort and post processing of course).

**Weaknesses:**

**Evaluation**
 - The significance of some results is unclear: Does the number of features in the initial prompt really have a significant effect on the performance, if the mean lies between [0.6823, 0.684] (Table 5). The same holds for the temperature metric in Table 4 [0.8169, 0.8197].
 - The approach is not compared against modern automatic feature interaction learning approaches. This could provide a upper bound on the performance:
    - DCN V2: Improved Deep & Cross Network and Practical Lessons for Web-scale Learning to Rank Systems
    - AutoInt: Automatic Feature Interaction Learning via Self-Attentive Neural Networks

**Questions:**

- How many responses on average are necessary to train a downstream model for classification and/or regression?
 - Did you consider using reflection and providing previous prompts as input to the model apart from in context examples?

---

> ### Author Response · Authors · 2024-11-30
> **Rebuttal by Authors**
>
> Dear Reviewer ogYM,
>
> We apologize for the delayed response. With the rebuttal period extended, we took additional time to thoroughly review and address your questions.
>
> Thank you for your insightful review.
> We are pleased to find that the soundness, replicability, and explainability of our approach were recognized as strengths.
> We want to address the highlighted weaknesses and answer the posed questions to improve our paper's quality and clarity.
> Please find below replies to your comments.
>
> > The significance of some results is unclear.
>
> We perform additional one-tailed paired t-tests to assess the statistical significance of the results presented in Tables 4 and 5.
> In **Table 4**, the performance improvement with the temperature at 1 compared to the temperature at 0.5 (0.8197 vs 0.8169) yields a p-value of less than 0.006.
> In **Table 5**, the performance improvement with 10 prompt examples compared to 1 prompt example (0.6840 vs 0.6823) yields a p-value of approximately 0.06.
> This shows that both the temperature and the number of example features in the prompt have a significant impact on the performance.
> We select the optimal temperature (1) and number of prompt examples (10) based on validation performance for achieving the best test results.
> We will include these significance results in the paper.
>
> In addition, we compute the mean percentage performance improvement of our method FEBP over the baseline methods with GPT-3.5 and GPT-4, respectively.
> We have added the new results to our revised version as **Tables 8 and 9 in New Section D.5**.
> To examine the performance difference between our method and the baselines when using Random Forests and LightGBM, we perform additional statistical tests excluding linear model results and add the new results to **Section D.7**.
> We have highlighted the relevant statistical test results in **Tables 14-16** for clarity.
>
> > How many responses on average are necessary to train a downstream model for classification and/or regression?
>
> From our experiments in **Section 5.3**, the average number of responses to train a downstream model
> is 362.7 with GPT-3.5 and 331.9 with GPT-4.
>
> We summarize the results as follows
>
> |         | GPT-3.5 | GPT-4 |
> |---------|---------|-------|
> | Linear  | 356.7   | 326.3 |
> | RF      | 370.4   | 330.0 |
> | LGBM    | 360.2   | 335.7 |
> | **Mean**    | 362.7   | 331.9 |
>
>
> > Did you consider using reflection and providing previous prompts as input to the model apart from in context examples?
>
> Thank your for the advice. We have experimented with providing previous prompts and LLM outputs as input to the LLM.
> However, we observe a degradation in performance as feature generation converges prematurely to suboptimal feature spaces.
> Additionally, the extra input significantly increases LLM generation costs.
> In contrast, our approach to providing feedback through in-context examples enables the LLM to explore a broader range of diverse features while remaining cost-efficient.

---

> > ### Author Response · Authors · 2024-12-02
> > **Followup on Rebuttal Phase Feedback**
> >
> > Dear Reviewer ogYM,
> >
> > Thank you for reviewing our paper and providing such valuable and constructive feedback. We have carefully studied your suggestions and made several revisions, adding extensive experimental details to enhance the paper’s clarity, depth, and contributions. Your insightful comments have been instrumental in guiding these improvements.
> >
> > As the rebuttal phase is nearing its deadline, we are looking forward to engaging in a timely discussion.
> > If you have any further questions or concerns, please do not hesitate to let us know. We are more than willing to provide additional clarifications and supporting materials to improve the quality of our work.
> >
> > Additionally, we kindly hope that these updates and clarifications will encourage you to reconsider your evaluation, as they directly address your constructive feedback.
> >
> > Thank you again for your invaluable time and effort!
> >
> > Best regards,
> >
> > Authors

---

### Official Review · Reviewer_B4Hm · 2024-11-05

**Soundness:** 2
**Presentation:** 3
**Contribution:** 2
**Rating:** 5
**Confidence:** 4

**Summary:**

The paper proposes an automated feature engineering solution by exploiting the data description to automatically generate new features. The idea is to leverage LLM and provide it with descriptive information of the dataset in the prompt to guide the search for effective features. The paper proposes a compact feature representation in the prompt and instructs the LLM to iteratively generate feature augmentation code that perform data transformation based on a set of pre-defined operators. Experimental evaluation demonstrates it effectiveness by comparing against the baselines.

**Strengths:**

S1. The problem of automated feature engineering is an important problem to solve given the significant effort a data scientist/analyst has to spend in a typical life cycle of data science workflow. The idea of leveraging LLM to interpret the descriptive information about the dataset and performing the appropriate data engineering operations can speed up the cycle.

S2. The paper is written with running examples. The core technical part is clear and easy to follow.

S3. The experimental evaluation includes comparison against the state-of-the-art technique. It demonstrates superior performance in some of the cases.

**Weaknesses:**

W1. While acknowledging the importance of the problem and the high level idea, the paper does not provide enough justification/discussion on its difference to CAAFE. CAAFE which is an existing work shares the same vision and solves the same problem by using LLM to iteratively augment the dataset to achieve better accuracy. What are the unique research challenges?

W2. The experimental evaluation does not show significant improvement against CAAFE.

W3. Adding some ablation studies will be appreciated. There lacks of experimental evaluations to support the design choice, e.g. cRPN.

**Questions:**

Q1. How is the vision and high level idea different from CAAFE? What are the unique research challenges undressed by CAAFE and are tackled here?

---

> ### Author Response · Authors · 2024-11-25
> **Rebuttal by Authors**
>
> Dear Reviewer B4Hm,
>
> Thank you for acknowledging the importance of our research problem and the clarity of the paper.
> We want to address the highlighted weaknesses and answer the posed questions to improve our paper's quality and clarity.
> Please find below replies to your comments.
>
> ---
> > W1. The paper does not provide enough justification/discussion on its difference to CAAFE.
>
> We propose a novel approach to LLM-based automated feature engineering
> that addresses the weaknesses of CAAFE.
> We add a discussion on the differences in **new Section G**.
> We highlight the key differences between our work FEBP and CAAFE as follows
> 1. In FEBP, we provide top-performing constructed features in the prompt as learning examples, label them with performance scores, and rank them by score.
> CAAFE instead stores all previous instructions and code snippets in the conversation history.
> The advantages of our approach include
>     - More effective **in-context learning of feature patterns**.
>     - The LLM generation cost **stays constant** across feature search iterations, without consuming extra LLM context.
>     - Stronger capability of performing feature search in **large search spaces** by enabling more iterations, providing pronounced benefits in presence of numerous feature attributes.
> 2. In FEBP, we represent features in the form of canonical RPN (cRPN), while CAAFE represents features with Python code.
> The advantages of our approach include
>     - cRPN is more compact, **reducing LLM generation costs**.
>     - The compactness of cRPN makes **in-context learning of feature patterns** easier.
>     - The use of pre-defined operators **reduces the search space** and **simplifies the learning process** for optimizing feature construction.
>     - Our approach gives **better control**, such as the number of operators to use in features, and helps avoid undesirable or unexpected LLM outputs, such as malicious code snippets.
>     - It is convenient to import external features and export the results as individual features, providing **compatibility with other AutoFE methods**.
> 3. We demonstrate in this work that general-purpose LLMs like GPTs can effectively model **recursive tree structures** in the form of cRPN feature expressions and reason about the structures in the context of **semantic information**.
>
> We hope this explanation makes the contributions of our work clearer.
>
> ---
> > W2. The experimental evaluation does not show significant improvement against CAAFE.
>
> In **Section D.7** (Section C.5 of the old version), we perform statistical tests to verify that the performance improvement of our method FEBP over CAAFE is statistically significant.
> The p-values from the Nemenyi post-hoc test are summarized as follows
>
> |          |         | **CAAFE**  |        |        |        |
> |----------|---------|------------|--------|--------|--------|
> |          |         | GPT-3.5    |        | GPT-4  |        |
> | **FEBP** | GPT-3.5 | **0.0016** | **0.0019** | **0.0010** | **0.0010** |
> |          |         | **0.0010** | **0.0010** | **0.0010** | **0.0010** |
> |          | GPT-4   | **0.0010** | **0.0010** | **0.0010** | **0.0010** |
> |          |         | **0.0105** | **0.0125** | **0.0010** | **0.0010** |
>
> We note that the performance improvement of our method FEBP is statistically significant at the p=0.01 level.
>
> To examine the performance difference when using Random Forests and LightGBM, we perform additional statistical tests excluding linear model results.
> The p-values from the Nemenyi post-hoc test are summarized as follows
>
> |      |         |  **CAAFE**  |        |        |        |
> |------|---------|:-------:|:------:|:------:|:------:|
> |      |         | GPT-3.5 |        |  GPT-4 |        |
> | **FEBP** | GPT-3.5 | 0.4493  | 0.3422 | **0.0010** | **0.0026** |
> |      |         | **0.0334**  | **0.0199** | **0.0010** | **0.0010** |
> |      |  GPT-4  | 0.1516  | 0.1017 | **0.0010** | **0.0010** |
> |      |         | 0.9000  | 0.9000 | **0.0299** | 0.0661 |
>
> We note that FEBP with GPT-3.5 and post-AutoFE parameter tuning significantly outperforms CAAFE in all cases at the p = 0.05 level.
> With GPT-4, the performance difference between FEBP and CAAFE is statistically significant at the p = 0.05 level.
>
> We have highlighted the corresponding parts of **Tables 14 and 16** in our revised version for better clarity.

---

> > ### Comment · Reviewer_B4Hm · 2024-11-26
> >
> > Thank you for your response which highlights the difference between FEBP and CAAFE. While acknowledging the improvement over CAAFE, the contribution seems incremental technically and a bit pale regarding the claim of being a noval method by having CAAFE as a prior art.

---

> ### Author Response · Authors · 2024-11-25
> **Rebuttal by Authors - Continued**
>
> > W3. Adding some ablation studies will be appreciated. There lacks of experimental evaluations to support the design choice, e.g. cRPN.
>
> Thank your for the advice. We add an additional ablation study regarding RPN canonicalization in **new Section E**.
> We summarize the new experimental results as follows
>
> |         | GPT-3.5 |        |       |            |        |       | GPT-4  |        |       |        |        |       |
> |---------|---------|--------|-------|------------|--------|-------|--------|--------|-------|--------|--------|-------|
> |         | w/o     |        |       | Full       |        |       | w/o    |        |       | Full   |        |       |
> | Linear  | 0.6471  | 0.6486 | 349.1 | **0.6485** | **0.6487** | 356.7 | 0.6462 | 0.6463 | 323.6 | **0.6532** | **0.6526** | 326.3 |
> | RF      | 0.7372  | 0.7373 | 358.0 | **0.7408** | **0.7412** | 370.4 | 0.7366 | 0.7366 | 315.7 | **0.7392** | **0.7393** | 333.0 |
> | LGBM    | 0.7485  | 0.7490 | 348.9 | **0.7522** | **0.7558** | 360.2 | 0.7461 | 0.7480 | 328.5 | **0.7542** | **0.7538** | 335.7 |
> | Mean    | 0.7141  | 0.7148 | 352.2 | **0.7171** | **0.7185** | 362.7 | 0.7128 | 0.7135 | 322.5 | **0.7187** | **0.7183** | 331.9 |
>
> **Without RPN canonicalization**, we observe performance degradation using all three downstream models.
> The Friedman-Nemenyi post-hoc test shows that the performance difference is statistically significant at the p=0.05 level
> for the cases with GPT-3.5 and post-AutoFE parameter tuning as well as GPT-4 without post-AutoFE parameter tuning.
> We also observe a decrease in the number of LLM responses without canonicalization. This is because the expression becomes more flexible, reducing the likelihood of duplication with existing features.
> Additionally, we find that when switching to **prefix feature expressions**, the LLM encounters difficulty generating syntactically valid feature expressions, leading to a failure to complete one single run in our experiments.
>
> Moreover, we add a discussion on why we use canonical RPN in **new Section A**.
> We represent features in **RPN** because it is compact and unambiguous and it better encodes the recursive structure for sequential modeling.
> We **canonicalize RPN** because that ensures the consistency of feature representations and facilitates the in-context learning of feature patterns.
> Our canonicalization scheme also introduces left skewness to the expression tree that enhances the clarity of the recursive structure in cRPN.
>
> Furthermore, we add an additional feature analysis in **new Section F.1**.
> We compare the proportions of constructed features selecting each feature attribute in the datasets for both the full and semantically blinded versions of FEBP.
> In the **blinded version**, we observe that the LLM tends to prioritize earlier feature attributes in the dataset while paying less attention to later ones, reflecting an inherent bias of the language model.
> In contrast, in the **full version**, the selection of feature attributes is guided by the semantic information of the dataset rather than the positional order of the attributes.
> This demonstrates the role of **dataset semantic information** in the LLM-based feature construction process.
>
> We also add an additional analysis of feature importance in **new Section F.2**.
> We find that the features constructed by our method have high importance in the downstream models.
> Moreover, our method adaptively adjusts the feature complexity to suit different downstream models.

---

> > ### Comment · Reviewer_B4Hm · 2024-11-26
> >
> > Thank you for your response to W3. This addresses my concern.

---

> ### Author Response · Authors · 2024-11-26
> **Followup on W2**
>
> Dear Reviewer B4Hm,
>
> Thank you very much for your response. We are glad that we have addressed your concern on W3.
> Next we provide additional information to address your concern on W2.
>
> > W2. The experimental evaluation does not show significant improvement against CAAFE.
>
> We compute the mean percentage performance improvement of our method FEBP over the baseline methods with GPT-3.5 and GPT-4, respectively.
> We add the full results to our revised version as **Tables 8 and 9 in New Section D.5**.
>
> With GPT-3.5,
>
> |        | Raw   |       | DIFER |       | OpenFE |       | CAAFE |       |
> |--------|-------|-------|-------|-------|--------|-------|-------|-------|
> | Linear | 22.17 | 22.33 | 4.76  | 2.36  | 14.30  | 14.50 | **12.35** | **12.50** |
> | RF     | 2.36  | 2.37  | 0.02  | -0.08 | 0.49   | 0.63  | **0.54**  | **0.67**  |
> | LGBM   | 2.60  | 3.16  | 0.18  | 0.32  | 1.21   | 1.58  | **0.82**  | **1.45**  |
> | Mean   | 8.39  | 8.63  | 1.50  | 0.79  | 4.88   | 5.12  | **4.18**  | **4.49**  |
>
> With GPT-4,
>
> |        | Raw   |       | DIFER |       | OpenFE |       | CAAFE |       |
> |--------|-------|-------|-------|-------|--------|-------|-------|-------|
> | Linear | 23.26 | 23.52 | 5.56  | 3.21  | 15.33  | 15.64 | **12.17** | **12.33** |
> | RF     | 2.08  | 2.07  | -0.26 | -0.37 | 0.21   | 0.33  | **0.69**  | **0.76**  |
> | LGBM   | 2.67  | 2.60  | 0.27  | -0.19 | 1.27   | 1.03  | **2.01**  | **1.15**  |
> | Mean   | 8.64  | 8.69  | 1.67  | 0.76  | 5.12   | 5.17  | **4.60**  | **4.37**  |
>
> We observe that our method has a significant amount of performance improvement over CAAFE.
> The greatest performance improvement over CAAFE is observed when using linear models.
>
> We would like to note that the reason why the performance improvement is less salient when using RF and LGBM
> is because these models can capture complex non-linear relationships from raw features themselves, which dilutes the effect of feature engineering.
> Thus, in our view, the performance improvement when using simple downstream models like linear models better demonstrates the efficacy of AutoFE.

---

> > ### Author Response · Authors · 2024-12-02
> > **Followup on Rebuttal Phase Feedback**
> >
> > Dear Reviewer B4Hm,
> >
> > Thank you for reviewing our paper and providing such valuable and constructive feedback. We have carefully studied your suggestions and made several revisions, adding extensive experimental details to enhance the paper’s clarity, depth, and contributions. Your insightful comments have been instrumental in guiding these improvements.
> >
> > As the rebuttal phase is nearing its deadline, we are looking forward to engaging in a timely discussion.
> > If you have any further questions or concerns, please do not hesitate to let us know. We are more than willing to provide additional clarifications and supporting materials to improve the quality of our work.
> >
> > Additionally, we kindly hope that these updates and clarifications will encourage you to reconsider your evaluation, as they directly address your constructive feedback.
> >
> > Thank you again for your invaluable time and effort!
> >
> > Best regards,
> >
> > Authors

---

### Author Response · Authors · 2024-11-29
**Global Response (Revision Summary)**

Dear Reviewers,

We thank all reviewers for the insightful questions and feedback. Your time and effort are sincerely appreciated.
In response to the reviewers' suggestions, we have made several key modifications in our revision, summarized as follows

1. **New Ssection A**: **New discussion** on why we use canonical RPN feature representation. [Reviewers `B4Hm`, `gWBg`, and `nSbg`]
2. **New Section D.5**: Percentage performance improvement of our method over the baseline methods. [Reviewers `B4Hm`, `ogYM`, and `nSbg`]
3. **Updated Section D.7**: Additional statistical test results excluding linear models. [Reviewers `B4Hm`, `ogYM`, and `nSbg`]
    - The performance difference between our method and all baselines other than DIFER is statistically significant at the p = 0.05 level.
4. **New Section E**: **Additional ablation study** on RPN canonicalization and prefix feature expressions.  [Reviewers `B4Hm`, `gWBg`, and `nSbg`]
    - The performance of our method without RPN canonicalization degrades. Our method fails when using prefix feature expressions.
5. **New Section F.1**: **Additional analysis** of the selection of feature attributes in the generated features. [Reviewers `B4Hm`, `gWBg`, and `nSbg`]
    - The dataset semantic information guides the selection of feature attributes, overcoming the bias of the positional order of feature attributes.
6. **New Section F.2**: **Additional analysis** of the feature importance. [Reviewers `B4Hm`, `gWBg`, and `nSbg`]
    - Our method adaptively adjusts the feature complexity to suit different downstream models.
7. **New Ssection G**: **New discussion** on the differences to CAAFE and the contributions of our work. [Reviewers `B4Hm`, `gWBg`, and `nSbg`]

We have also improved the overall clarity of the revised paper and corrected all typos identified.

We hope these revisions address the reviewers' concerns and improve the overall quality of our paper.

Thank you again for your review!

Best regards,

Authors

---

> ### Author Response · Authors · 2024-12-04
> **Global Response**
>
> Dear Reviewers,
>
> Thank you again for your review. We summarize the key points in our responses
>
> **Reviewer B4Hm**
>
> - We explain the differences between our work and CAAFE.
>   Our approach addresses the weaknesses of CAAFE by adopting compact feature representations and providing example
>   features in the prompt, leading to stronger feature search performance.
> - We compute the relative performance improvement of our method over CAAFE and perform statistical tests
>   to show that the performance improvement of our method over CAAFE is significant.
> - We conduct an additional ablation study on RPN canonicalization. We add additional explanations on why we use
>   canonical RPN for feature representation.
>
> **Reviewer ogYM**
>
> - Statistical tests show that the performance differences in our studies on the LLM temperature and the number of
>   example features are significant.
> - We explain the number of LLM responses on average to train a downstream model.
> - We find that providing previous prompts to the LLM would lead to premature convergence to suboptimal feature spaces.
>   In contrast, our approach enables the LLM to explore a broader range of diverse features while remaining
>   cost-efficient.
>
> **Reviewer gWBg**
>
> - AutoFE suits small datasets and provides controllability of the feature generation process and the interpretability of
>   the extracted features,
>   which requires a give set of operators.
> - We propose a novel approach to LLM-based automated feature engineering that addresses the limitations of the
>   state-of-the-art approach.
> - We conduct an additional ablation study on RPN canonicalization and perform additional analyses on the selection of
>   feature attributes and the feature importance.
> - Temperature is a relevant hyperparameter that impacts both the performance and the feature construction efficiency of
>   our method.
> - The performance with GPT-4 improves as the number of example features increases.
>   Incorporating more example features helps fully leverage GPT-4's enhanced in-context
>   learning capabilities.
>
> **Reviewer nSbg**
>
> - We propose a novel approach to LLM-based automated feature engineering that addresses the limitations of the
>   state-of-the-art approach.
> - We conduct an additional ablation study on RPN canonicalization and perform additional analyses on the selection of
>   feature attributes and the feature importance.
> - The relative performance improvement of our method over the baselines and additional statistical tests show that there
>   are significant differences in performance
>   when using Light-GBM and random forests.
>
> In this work, we introduce a novel LLM-based AutoFE approach that addresses the limitations of the
> state-of-the-art method.
> Through extensive experiments, we demonstrate that our approach achieves superior performance compared to
> state-of-the-art baselines.
> We provide detailed analyses on the impact of dataset semantic information, RPN canonicalization, and
> hyperparameter settings.
> Additionally, we present details of the LLM-based feature search process, the selection of feature attributes, and the
> evaluation of feature importance.
> We believe our work makes valuable contributions to the fields of automated feature engineering and
> LLM-driven applications.
> We respectfully request your consideration of our paper for acceptance.
> Thank you for your time and thoughtful review.
>
> Best regards,
>
> Authors

---

### Meta-Review · Area_Chair_obTY · 2024-12-16

**Metareview:**

This paper proposes an automatic feature engineering framework leveraging large language models (LLMs). The framework uses dataset metadata and feature relevance as input to sequentially generate new features. While the experimental results demonstrate improved performance compared to alternative methods, I have concerns regarding the significance of its contributions.

The paper is well-written and presents a clear, reproducible methodology. The authors conduct extensive experiments across various datasets and compare their approach with multiple existing methods. However, I concur with the other reviewers that the novelty of this work is limited. Despite the authors' attempts to clarify the novel aspects of their framework, the contributions appear incremental and do not directly address the limitations of previous approaches.

Furthermore, the performance gains are marginal, especially when evaluated with more complex downstream models. Additionally, some experimental results, particularly the comparison between GPT-3.5 and GPT-4, are counterintuitive and require further investigation.  Due to these weaknesses, I am rejecting this paper.

**Additional Comments On Reviewer Discussion:**

I appreciate the authors' efforts to address the reviewers' concerns during the rebuttal period. They clarified the contributions and novelties of their proposed method compared to existing alternatives and provided additional experiments to demonstrate the statistical significance of the performance improvements.

However, despite these efforts, some concerns remain. The technical novelty compared to previous methods still requires further clarification, and the performance improvements, while statistically significant, are somewhat limited. Additionally, inconsistencies in the experimental results persist and need to be addressed.

By focusing on these remaining issues, particularly by highlighting the unique technical contributions and ensuring consistent and robust experimental results, the authors could significantly strengthen this paper.

---

### Decision · Program_Chairs · 2025-01-22

Reject